# Immunodominant proteins P1 and P40/P90 from human pathogen *Mycoplasma pneumoniae*

David Vizarraga [1], Akihiro Kawamoto [2,3], U. Matsumoto[4], Ramiro Illanes [1], Rosa Pérez-Luque[1], Jesús Martín[1], Rocco Mazzolini[5], Paula Bierge[6], Oscar Q. Pich [6,7], Mateu Espasa[8], Isabel Sanfeliu[8], Juliana Esperalba[9], Miguel Fernández-Huerta[9], Margot P. Scheffer[10], Jaume Pinyol[7], Achilleas S. Frangakis [10], Maria Lluch-Senar[5], Shigetarou Mori [11], Keigo Shibayama[11], Tsuyoshi Kenri [11], Takayuki Kato [2,3], Keiichi Namba [2,12,13], Ignacio Fita[1], Makoto Miyata [4,14✉] & David Aparicio [1✉]

*Mycoplasma pneumoniae* is a bacterial human pathogen that causes primary atypical pneumonia. *M. pneumoniae* motility and infectivity are mediated by the immunodominant proteins P1 and P40/P90, which form a transmembrane adhesion complex. Here we report the structure of P1, determined by X-ray crystallography and cryo-electron microscopy, and the X-ray structure of P40/P90. Contrary to what had been suggested, the binding site for sialic acid was found in P40/P90 and not in P1. Genetic and clinical variability concentrates on the N-terminal domain surfaces of P1 and P40/P90. Polyclonal antibodies generated against the mostly conserved C-terminal domain of P1 inhibited adhesion of *M. pneumoniae*, and serology assays with sera from infected patients were positive when tested against this C-terminal domain. P40/P90 also showed strong reactivity against human infected sera. The architectural elements determined for P1 and P40/P90 open new possibilities in vaccine development against *M. pneumoniae* infections.

[1] Instituto de Biología Molecular de Barcelona (IBMB-CSIC), Parc Científic de Barcelona, Baldiri Reixac 10, 08028 Barcelona, Spain. [2] Graduate School of Frontier Biosciences, Osaka University, 1-3 Yamadaoka, Suita, Osaka 565-0871, Japan. [3] Institute for Protein Research, Osaka University, 3-2 Yamadaoka, Suita, Osaka 565-0871, Japan. [4] Graduate School of Science, Osaka City University, Osaka 558-8585, Japan. [5] EMBL/CRG Systems Biology Research Unit, Centre for Genomic Regulation (CRG), The Barcelona Institute of Science and Technology, Dr Aiguader 88, 08003 Barcelona, Spain. [6] Laboratori de Recerca en Microbiologia i Malalties Infeccioses, Institut d'Investigació i Innovació Parc Taulí (I3PT), Hospital Universitari Parc Taulí, Universitat Autònoma de Barcelona, 08208 Sabadell, Spain. [7] Departament de Bioquímica i Biologia Molecular, Institut de Biotecnologia i Biomedicina, Universitat Autònoma de Barcelona, 08193 Bellaterra, Barcelona, Spain. [8] Departament de Microbiologia, Hospital Universitari Parc Taulí, Universitat Autònoma de Barcelona, 08208 Sabadell, Spain. [9] Departament de Microbiologia, Hospital Universitari Vall d´Hebron, Universitat Autònoma de Barcelona, 08035 Barcelona, Spain. [10] Buchmann Institute for Molecular Life Sciences, Max-von-Laue Str. 15, 60438 Frankfurt, Germany. [11] Department of Bacteriology II, National Institute of Infectious Diseases, Tokyo, Japan. [12] RIKEN Center for Biosystems Dynamics Research and SPring-8 Center, 1-3 Yamadaoka, Suita, Osaka 565-0871, Japan. [13] JEOL YOKOGUSHI Research Alliance Laboratories, Osaka University, 1-3 Yamadaoka, Suita, Osaka 565-0871, Japan. [14] The OCU Advanced Research Institute for Natural Science and Technology (OCARINA), Osaka City University, Osaka 558-8585, Japan. ✉email: miyata@sci.osaka-cu.ac.jp; daacri@ibmb.csic.es

M ycoplasma pneumoniae is a human pathogen responsible for upper and lower respiratory tract infections[1]. It is estimated that this bacterium is responsible for up to 40% of community-acquired pneumonias in persons of all ages[2]. In addition to being a severe respiratory pathogen, *M. pneumoniae* may induce clinically significant manifestations in extrapulmonary sites and/or immunologic effects in as many as 25% of the infections[3]. Unlike for other important respiratory pathogens, such as *Streptococcus pneumoniae* and *Haemophilus*

*influenzae*, a vaccine for *M. pneumoniae* is not yet available despite the considerable efforts[4].

*M. pneumoniae* binds to host target cells by means of a polar structure known as the attachment organelle, and glides in the direction of this differentiated structure at a speed of ~1 µm/s (Fig. 1a)[5]. Gliding motility in *M. pneumoniae* and related species such as *Mycoplasma genitalium* is caused by a unique mechanism completely different from the motility of *Mycoplasma mobile*, which has been studied in further detail[6–8]. Adherence of *M.*

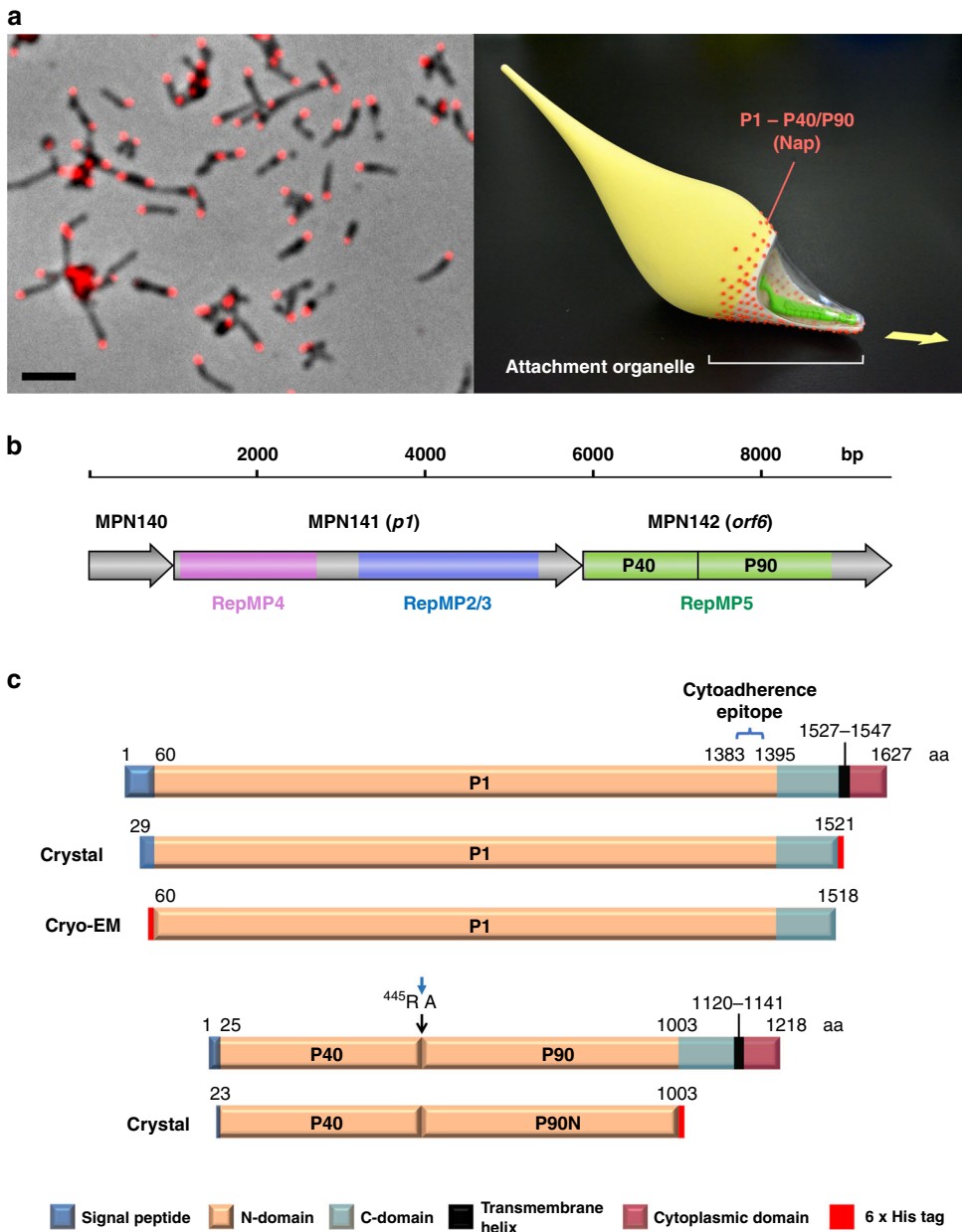

**Fig. 1 The Nap complex in *M. pneumoniae*. a** A phase-contrast immunofluorescence microscopy image of *M. pneumoniae* cells (left). Locations of the Nap (surface of the attachment organelle) were visualized by using anti-P1 monoclonal antibody and Alexa Fluor® 546 secondary antibody (red). Bar = 2 µm. Three-dimensionally printed model of *M. pneumoniae* cell (right). The positions of the attachment organelle and the Nap structure are indicated. The cell surface is partly modeled as a transparent canopy to show the internal core structure of the organelle. The arrow indicates gliding direction. **b** Scheme of the *p1* operon that contains three genes: MPN140, MPN141(*p1*), and MPN142(*orf6*). The locations of repetitive elements (RepMPs) in MPN141 and MPN142 are shown by colors. The regions corresponding to P40 and P90 are also shown in MPN142. **c** Scheme of native P1 and P40/P90 proteins and the recombinant constructs used for crystallization and cryo-EM analysis. The major cytoadherence epitope of P1 reported previously is indicated (1383–1395). The boundary between P40 and P90N proteins identified in this study (445 R/A) is indicated by arrows (Supplementary Fig. 5). Positions of 6 × His-tag used for purification of recombinants are shown by red boxes.

*pneumoniae* to cells of the respiratory tract is mediated by a network of adhesins and cytoadherence accessory proteins[9,10]. Within this network the 170 kDa protein P1 was identified as a major determinant for cytoadherence and gliding motility with antibodies against P1 preventing both adhesion and motility[9–12]. For about 40 years, P1 has been assumed to be responsible for binding to sialic acid oligosaccharide receptors from the host cells[13]. Because all these relevant properties, P1 has been attracting attention since the late 1970s, although it was soon recognized that accessory proteins were also required for its functioning[14]. P1, together with the P40/P90 polypeptides, forms a transmembrane complex called the "Nap"[15,16]. The structure of the Nap complex of *M. genitalium*, a human urogenital pathogen closely related to *M. pneumoniae*, was recently revealed by single-particle cryo-electron microscopy (cryo-EM) at 9 Å and cryo-electron tomography (cryo-ET) at ~15 Å resolution[17]. The *M. genitalium* Nap structure consists of a dimer of a P140–P110 complex protruding outward from the mycoplasma membrane and forming a large knob with a diameter of ~15 nm. P140 and P110 are the *M. genitalium* orthologues of P1 and P40/P90, respectively. Therefore, it was most unexpected to find that in *M. genitalium* the binding site for sialic acid oligosaccharides is in P110 and not in P140[17,18]. P1 is considered to be one of the most immunodominant proteins in *M. pneumoniae* cells, playing a major role in the immune response of infected patients and accordingly in diagnosis and epidemiologic studies[3,19,20]. The *M. pneumoniae* genome contains repeated regions, denominated RepMPs. The majority (75%) of RepMPs has homology with MPN141 (P1) and MPN142 (P40/P90) (Fig. 1b). Homologous recombination between RepMPs and either MPN141 or MPN142 generate variability within antigenic regions of P1 and P40/P90, respectively, providing an essential strategy to evade the immune host system[21–23]. P40/P90 consists of two polypeptides from the proteolytic cleavage of a 130 kDa translate encoded by MPN142[24] (Fig. 1b). Historically little attention has been dedicated to P40/P90 and the contribution of P40/P90 to cytoadherence is much less studied than for P1. Even the presence of the P40 polypeptide in the Nap has been questioned and the reason for and mechanism of the MPN142 cleavage remain unknown[25].

In the present work, we report the structures of *M. pneumoniae* proteins P1 and P40/P90 and of complexes of P40/P90 bound to sialylated oligosaccharides. The results elucidate the antigenic organization of these immunodominant proteins allowing the mapping of the genetic and clinical variability. Moreover, serological tests on different fragments of P1 reveal the recognition preference of IgG antibodies for conserved over variables regions, suggesting an immune system strategy of selecting these regions for long-term immunity. The information is considered pivotal for clarifying the motility, infectivity, and epidemics of *M. pneumoniae*.

## Results

**Structure determination of P1**. Crystals were obtained of the *M. pneumoniae* Nap protein P1 extracellular region (residues Thr29-Asp1521). This P1 construct, empirically chosen after many crystallization attempts, excludes the predicted N-end of the signal secretion peptide (spanning residue Met1 to Leu59 according to the structural properties prediction program Psipred[26]). In this construct are also excluded the (predicted) transmembrane and intracellular regions at the C-end of P1 (residues Tyr1522 to Ala1627) (Fig. 1c). These P1 crystals were solved simultaneously with the determination of the crystal structure from the *M. genitalium* orthologous protein P140, by averaging between crystals from both proteins (with 41% sequence identity) despite neither molecular models nor experimental phases were available (see "Methods") (Supplementary Fig. 1). The final refined model of P1, at 1.9 Å

resolution, has agreement $R$ and $R_{\mathrm{free}}$ factors of 18.7 and 22.9, respectively (Fig. 2a, b and Supplementary Table 1).

The structure of the ectodomain from P1 was also studied by single-particle analysis using cryo-EM (Figs. 1c and 2b and Supplementary Fig. 2). Although P1 particles were clearly observed in the cryo-EM micrographs, the quality of the map was too poor to refine the atomic model owing to severe orientation bias. To alleviate this problem, we collected data by using grids covered with a thin film of graphene oxide (GO), which drastically improved the data quality[27] (Supplementary Fig. 2a). A total of 68,014 selected particles yielded a 3D EM map at an overall resolution of ~2.9 Å, according to the gold-standard Fourier shell correlation (FSC) 0.143 criterion (see "Methods") (Supplementary Table 2).

The structure of P1 consists of a large N-terminal domain, with Asn60 as the first residue visible in the X-ray and the cryo-EM maps, and a smaller C-terminal domain (residues Gly1395–Asp1521) (Fig. 2). The N-terminal domain of P1 has a β-sheet propeller topology of seven consecutive blades or β-sheets, each with four antiparallel strands, except for β-sheet VII that contains only two strands. Interestingly, these two strands correspond exactly to a peptide with important immune properties, which had been defined as the cytoadherence epitope[28] (Fig. 2c). The lengths of connections between the β-sheets of the propeller are very diverse, ranging from just seven residues between sheets VI and VII, to 315 residues between sheets IV and V. Links between adjacent β-strands in the β-sheets of the propeller, also vary widely in length, ranging from just 11 to 219 residues. The longest intra and inter β-sheet connections cluster together creating a crown-like structure on one side of the β-propeller opposite to the position where the C-terminal domain is located. In the crown there are 13 disordered loops, spanning 117 residues, which could not be built (Fig. 2c). These disordered regions, essentially the same in the X-ray and cryo-EM structures, are spread over the N-terminal domain surface (Supplementary Figs. 1 and 3a). Therefore, the N-terminal domain of P1 can be described as a structured core surrounded by a fuzzy surface. The C-terminal domain contains only one disordered loop (residues Thr1483–Asn1495) and has an elongated fold with the same unique topology (according to DALI[29]) as the C-terminal domain of adhesin P110 from *M. genitalium*.

The cryo-EM and X-ray maps of P1 are well defined for most of the N-terminal domain with the residues clearly identifiable, in general, in both maps (Fig. 2b). However, from residue Pro1388 to the C-end, a region that includes the hinge between domains and the whole C-terminal domain, the density in the cryo-EM map becomes weak, indicating that there are hinge-like movements between domains. In the crystal structure, the C-terminal domain is anchored by crystal contacts and the density is clear and crisp, while in the top part of the crown, with few crystal contacts, temperature factors are high (Supplementary Fig. 3b) and chain tracing was difficult. Together the cryo-EM and X-ray maps provide accurate structural information about the whole P1 ectodomain, proving the mobility of domains with respect to one another and confirming the characteristic presence of a large number of disordered loops.

**Crystal structure of P40/P90N**. Diffracting crystals could not be obtained with constructs of the complete extracellular region of P40/P90, which is predicted by Psipred[26] to span from Ala25 to Pro1113 (with Met1 to Leu24 corresponding to a signal N-peptide) (Fig. 1c). After numerous attempts, high-resolution diffracting crystals were grown from a construct, referred henceforth as P40/P90N, spanning residues Ser23-Val1003 that, by sequence alignment, corresponds to the N-terminal domain of the *M.*

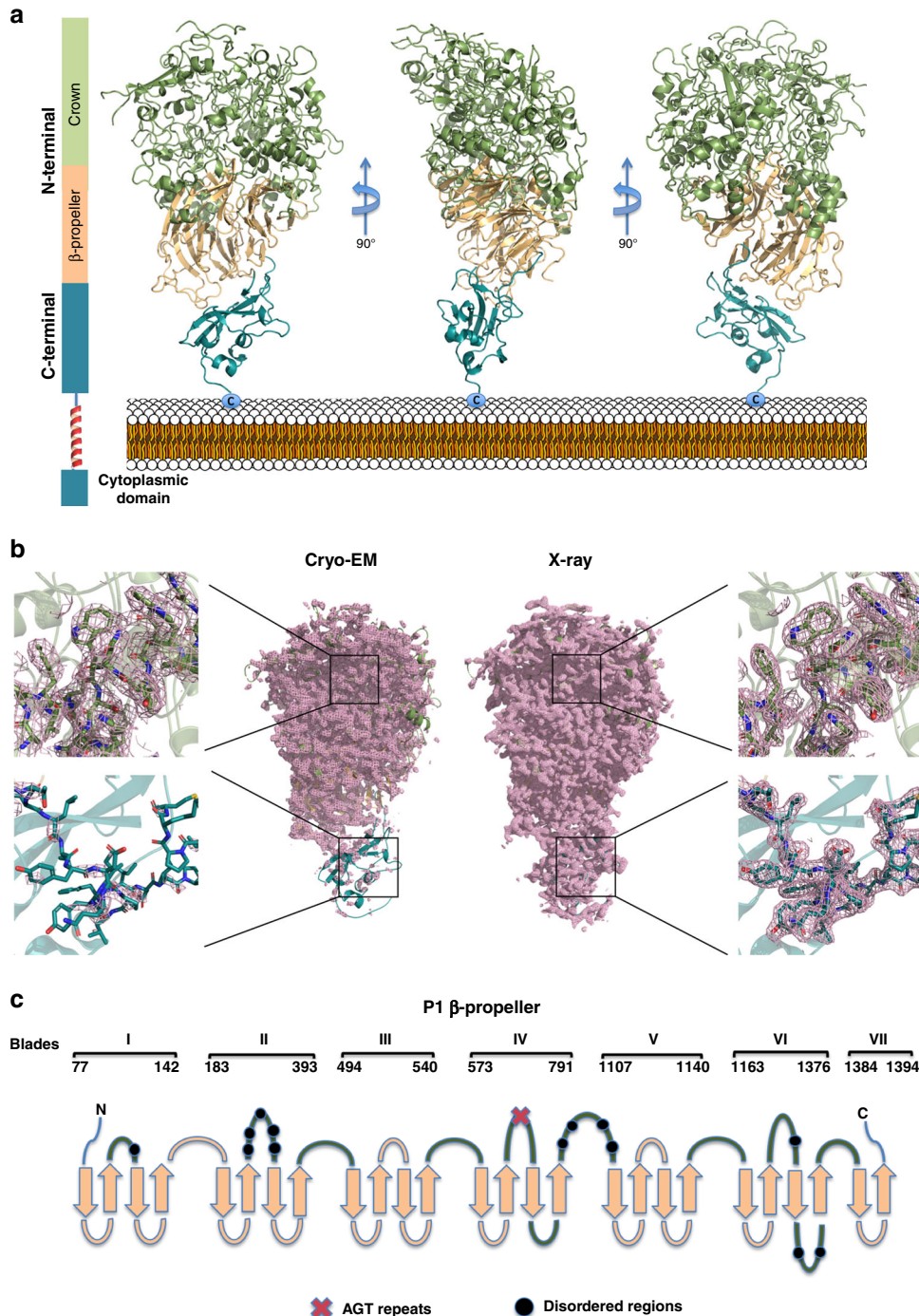

**Fig. 2 X-ray and cryo-EM structures of P1. a** Ribbon representation, with three 90° apart views, of the complete extracellular region of P1. The large N-terminal domain is organized around a seven-blade β-propeller (light orange) from where the crown (green) emerges. The C-terminal domain (cyan) is immediately followed by a predicted transmembrane helix, which requires the proximity of this domain to the cell membrane. **b** P1 map densities from cryo-EM (left panels) and X-ray crystallography (right panels). Insets show representative regions from the N-terminal domain (top) and the C-terminal domain (bottom). The quality of the cryo-EM density provides the clear identification of residues in the N-terminal domain (at an estimated resolution of 2.9 Å). On the contrary, the C-terminal domain is poorly defined in the cryo-EM map, indicating high flexibility with respect to the N-terminal domain. The core of the N-terminal domain and, especially, the C-terminal domain are well defined in the X-ray map, while the very top of the N-terminal domain presents high-temperature factors and was difficult to trace. **c** Topology of the β-propeller in P1. Loops contributing to the crown are colored green. Disordered loops are indicated with black dots and the stretch of serines (the AGT repeats) with a red cross. Strands and loops are not in scale.

*genitalium* orthologous protein P110 (Supplementary Fig. 4). The P40/P90N crystals, with two subunits in the asymmetric unit, were solved by molecular replacement using as a searching model the structure of P110 (PDB entry 6R3T), with a sequence identity between the extracellular regions of 44%. The refined P40/P90N

structure, at 2.65 Å resolution, has agreement $R$ and $R_{\text{free}}$ factors of 21.4 and 23.4, respectively (Supplementary Table 1).

The structure of P40/P90N reveals a large globular domain that is structurally closely related to the N-terminal domain of P110, with a RMSD of 1.2 Å for 742 equivalent residues[18] (Fig. 3a and

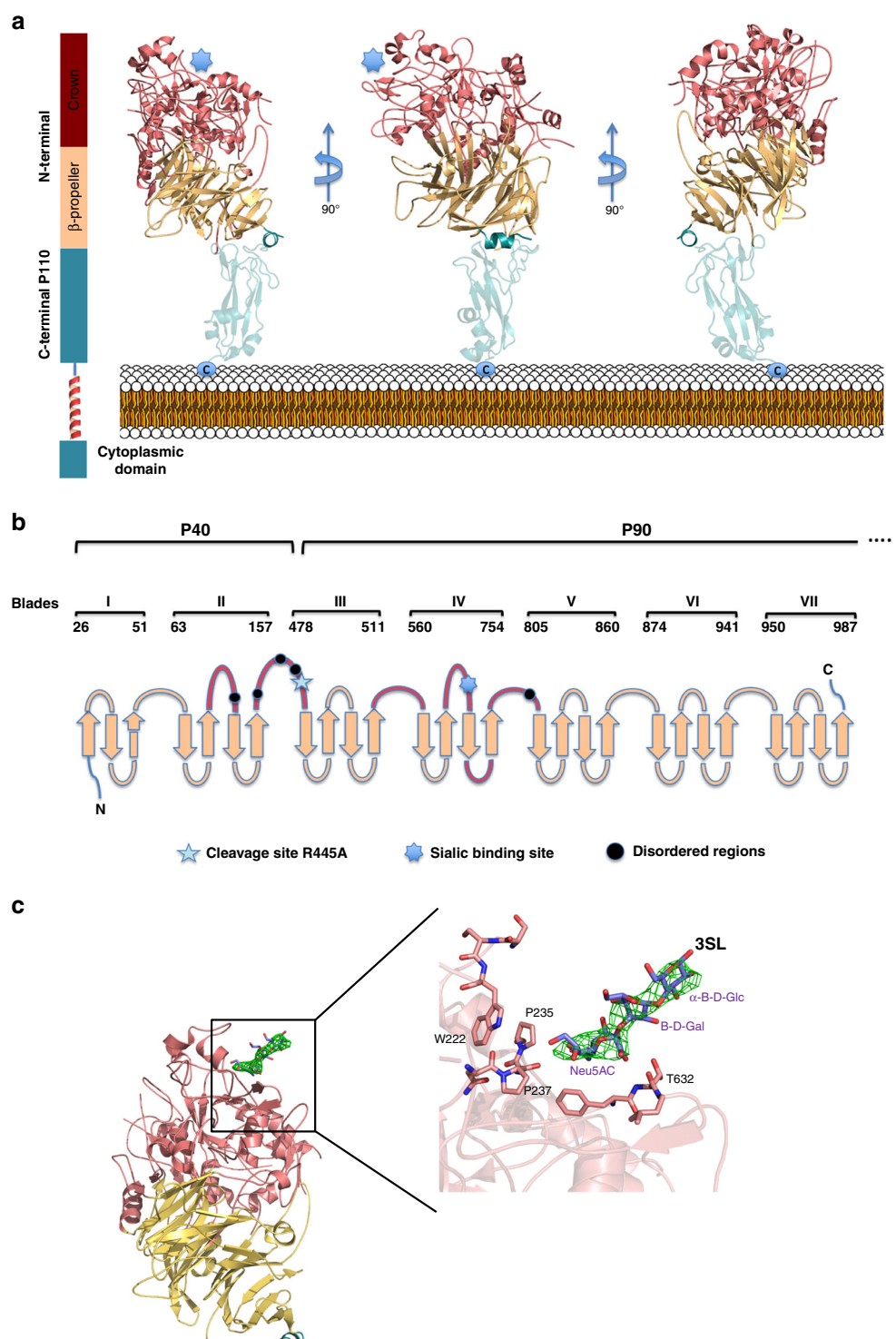

**Fig. 3 X-ray structure of P40/P90 and of its complex with oligosaccharide 3SL. a** Ribbon representation, with three 90° apart views, of the crystal structure from the P40/P90 N-terminal domain that is organized, similarly to P1, around a seven-blade β-propeller (light orange) from where it emerges the crown (pink). The sialic binding site found in P40/P90 is explicitly indicated. The predicted C-terminal domain of P40/P90 is also shown (cyan), modeled from the C-terminal domain of the orthologous adhesin P110 from *M. genitalium*. **b** Topology of the β-propeller from P40/P90. Loops contributing to the crown are colored pink. The first blade β-sheet contains only three strands in P40/P90 placing the N-end at the propeller face opposite to the one in P1. Localizations of the disordered loops and sialic binding site are indicated with black dots (black) and a star (blue), respectively. The position of the cleavage site, which results in the corresponding P40 and P90 separate polypeptides, is depicted with a pentagram (pale blue). Strands and loops are not in scale. **c** Binding of 3SL to the N-terminal domain of P40/P90. Ribbon representation of the P40/P90N structure in complex with oligosaccharide 3SL. The inset shows a detail of the residues shaping the binding pocket. The electron density omit map (at 0.9 sigma) corresponding to 3SL is also shown.

Supplementary Fig. 4). The sequence of P40/P90N presents two long insertions with respect to P110 of more than 60 residues each (Gly297–Thr368 and Leu395–Thr462 that are referred as S1 and S2, respectively) (Fig. 3b and Supplementary Fig. 4). Both insertions, completely disordered in the crystal structures determined, are found between β-propeller blades II and III. P40/P90, like P1, has a seven-blade β-propeller topology with many of the connections between the β-strands clustering together and forming the crown (Fig. 3b).

P40 and P90 had often been considered as two independent proteins resulting from the posttranslational processing of the MPN142 gene product[25,30]. In our hands, the purified P40/P90 samples present a characteristic SDS-PAGE gel pattern indicating a slow progression of cleavage in a few specific locations (Supplementary Fig. 5a). The primary scission occurs between Arg445 and Ala446, located toward the C-end of insertion S2 (Fig. 3b and Supplementary Figs. 4 and 5a). The two polypeptides produced by this primary cleavage, the shorter spanning from the N-end to Arg445 and the longer from Ala446 to the C-end, can correspond to the subunits P40 and P90 that are found in mycoplasma cells[30]. When the primary scission site was mutated to avoid cleavage, the P40/P90 variant protein (Ser445–Ser446) presented, unexpectedly, a similar proteolysis pattern with cleavage taking place between Arg455 and Ala456, still in the S2 insertion (Supplementary Fig. 5a). Studies with mycoplasma cells had reported two cleavage sites occurring between Arg445–Ala446[30] and (probably) Leu454–Arg455[25,31], in good agreement with the two cleavage Arg–Ala sites (RA motifs) now found in vitro. Regardless of the extend of proteolysis, all P40/P90N samples behave as a single and sharp peak by SEC–MALS (Supplementary Fig. 5b) indicating that cleavage takes place only when folding is completed and that a unique three-dimensional structure of P40/P90N is maintained. Structural stability of the cleaved protein probably relies on the large interface between the P40 and P90 polypeptides, with 75 hydrogen bonds, 6 salt bridges, and a buried surface area of 10,480 Å$^2$, which gives an estimated free energy of −55 kcal with an expected 100% probability of formation according to the Complexation Significance Score algorithm in PISA[32] (Supplementary Fig. 6a). Addition of EDTA to the P40/P90 samples halts proteolysis completely, which suggests a key role for divalent metals. No metals were found in the crystal structures of P40/P90, but EDTA was required for the crystallization conditions. The presence of significant amounts of Zn$^{2+}$ ions in a purified P40/P90 sample was detected by inductive coupled plasma–mass spectrometry (ICP–MS), although the concentration of protein was higher than the one found for the ion (see "Methods"). The possible presence in the P40/P90 samples of an undetected protease, which could be coming from the over-expression in Escherichia coli, was analyzed by SEC–MALS. Different constructs of P40/P90N, with and without the C-terminal domain, always produced a sharp peak at the molecular weight corresponding to the construct, without any indication of other species. Altogether these results suggest that cleavage of P40/P90 is a self-proteolytic process. However, we have not yet been able to find the residues that could be responsible for this enzymatic activity. Besides insertions S1 and S2, the structure of P40/P90 presents only three short disordered loops (residues Ser118–Gly127, Ser168–Gly175, and Ala774–Arg777) and the N-terminal domain from P40/P90 can be seen as a well-defined globular structure with two long floppy pendants (Supplementary Fig. 3a). The complete extracellular region of P40/P90 was generated by combining the structure of the N-terminal domain with the structure of the Pro1004–Pro1113 region of P40/P90 modeled according to the P110 C-terminal domain because the sequence identity is high (68%) and the expected RMSD low (~0.6 Å)[33] (Fig. 3a and Supplementary Fig. 4). In solution, P1 and P40/

P90 form an heterodimer showing a single peak in a gel filtration profile (Supplementary Fig. 5c).

**Complexes of P40/P90N with sialylated oligosaccharides.** The presence of a sialic binding site in P40/P90 was confirmed by the crystal structures of complexes of P40/P90N with oligosaccharides 3SL (neuraminic acid forming an α2–3 linkage to a lactose monosaccharide) and 6SL (with an α2–6 linkage) (Supplementary Table 1). This result is consistent with the facts that sialylated oligosaccharides act as cell receptors for M. pneumoniae and M. genitalium[34–36], and that adhesin P110 from M. genitalium, the orthologue of P40/P90 contains a binding site specific for neuraminic acid[18]. The sialic binding site of P40/P90 is in a location structurally equivalent to the one in P110, in the upper part of the crown (Fig. 3a, c and Supplementary Fig. 7). Only the neuraminic acid moiety interacts directly with the P40/P90 β-hairpin that contains the X-Tyr/Phe-Ser/Thr motif (Leu630–Phe631–Thr632) characteristic for the binding to sialic compounds[18,34]. P40 functions as the external rim of the sialic binding site pocket, which would become exposed without P40 (Supplementary Fig. 6b). Surface plasmon resonance (SPR) indicates that sialylated oligosaccharides bind to P40/P90 with dissociation constants ($K_d$) in the micromolar range for 3SL (~8.0) and 6SL (~2.9), similar to the values reported for P110[18] (Supplementary Fig. 8) although, according to in vivo studies, 6SL binds at a much lower affinity than 3SL[37,38]. For P1, SPR shows no binding to sialylated oligosaccharides, in agreement with what is observed for M. genitalium[17].

**Structural relationships of P1 and P40/P90.** The structural results reveal that P1 (1627 residues) and P40/P90 (1218 residues) present a similar overall organization and topology in the extracellular regions. The similarity is maintained between the transmembrane helices and the cytoplasmic regions where the sequence identity of P1 and P40/P90 is high (43%). Despite the different size, the close relationship of both proteins suggests that P1 and P40/P90 have a recent common phylogenetic ancestor. Superposition of N-terminal domains from P1 and P40/P90, with a sequence identity of only 12%, gives a RMSD of 3.5 Å for 361 equivalent residues, which correspond mainly to residues from the β-sheets of the propellers. However, β-sheet I has only three strands in P40/P90, but four in P1 and this "extra" first strand in the first β-sheet of P1 places the N-end in the propeller face that is the opposite to the one in P40/P90 (Fig. 4). The "extra" first strand of β-sheet I from P1 starts with a large β-bulge (residues 69–77) occluding the center of the propeller, contrary to P40/P90 where the center is empty. Linkage between the N- and C-terminal domains of P1 changes with respect to P40/P90, due to the absence of the last two strands in β-sheet VII. Therefore, β-sheets I and VII, structurally contiguous within the propeller, present the largest differences between P1 and P40/P90. The hypothesis of a recent common origin is further supported by the close structural relationships between the C-terminal domains from the non-orthologous proteins P1 and P110, with a sequence identity of 19% and a RMSD of 2.0 Å for 73 equivalent residues (Supplementary Fig. 9a). The hypothesis is also reinforced by the presence of the characteristic "AGT repeats" motif in the same loop of the propellers, between the second and third β-strands of the fourth blade, of P1 and P110 (Supplementary Fig. 9b).

**Mapping of epitopes and genetic variability onto P1 and P40/P90.** Epitopes reported for six monoclonal antibodies (mAbs) against P1 with inhibiting cytoadherence activities for M. pneumoniae (aN, P1.62, P1.26, M51, M58, and 6E7)[28,39] (Supplementary Fig. 10) are all exposed on the surface of P1, as expected

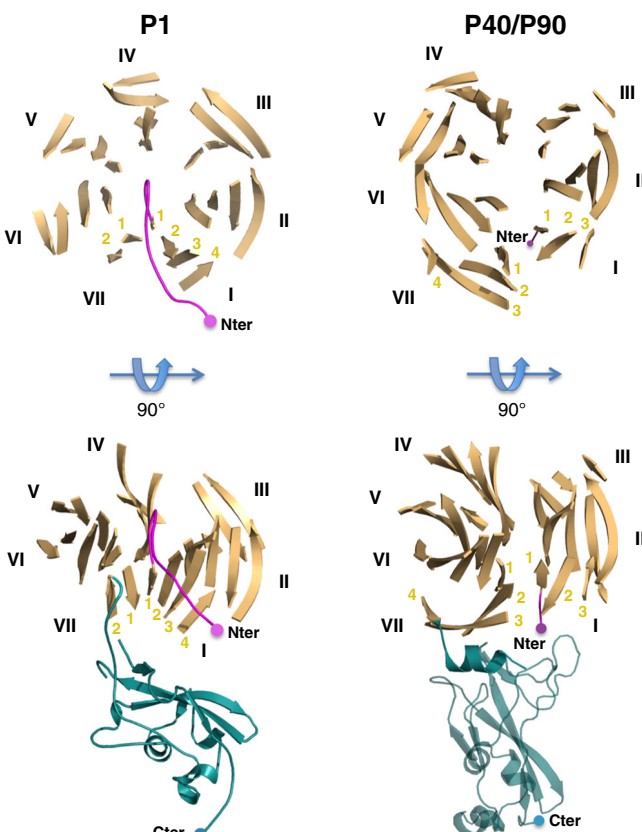

**Fig. 4 Comparison of P1 and P40/P90.** Comparison of the β-propeller structures and of their connections with the corresponding C-terminal domains in P1 (left panels) and P40/P90 (right panels). The C-terminal domain is tangential to the propeller ring in P1 and protrudes radially in P40/P90, where the C-terminal domain was modeled from the orthologous adhesin P110 from *M. genitalium*. The N-ends are situated in opposite faces of the propeller in P1 and P40/P90. The center of the propeller is occluded in P1 but empty in P40/P90.

to be accessible by the antibodies (Fig. 5a). Epitopes from mAbs aN, P1.62, P1.26, and 6E7 are located in the N-terminal domain, whereas epitopes from mAbs M51 and M58 are in the C-terminal domain. The epitope from mAb 6E7 corresponds to the strands from the last β-sheet of the β-propeller, close to the C-terminal domain. The mAbs, P1.62 and P1.26 are reported to bind to dual epitopes (Supplementary Fig. 10), separated on the structure of P1 (Fig. 5a). Although dual epitopes of P1.26 are both exposed in the structure of P1, one of them positioned on Asn980–Lys987 might become hidden in the Nap complex.

Two other reported epitopes (Trp810–Tyr817 and Phe1124–Arg1131), recognized by sera from many of the *M. pneumoniae* infected patients[20], are located in the N-terminal domain (Fig. 5a). The first of these exist inside the P1 molecule, while the second is exposed in the P1 surface, although its accessibility might also be limited in the Nap complex. These two epitopes might be immunodominant for antibody production; however, antibodies against these epitopes may not have cytoadherence-inhibitory activities.

Sequences of P1 and P40/P90 exhibit sequence variations depending on the *M. pneumoniae* strain. The two major variation types, known as 1 and 2, are harbored in strains M129 and FH, respectively[22,31,40]. For P1 the main source of variability is RepMP4 and RepMP2/3, while for P40/P90 is RepMP5[41]. It has been reported that mAb P1.62 only binds to type 2 P1 protein[42],

consistent with the fact that the epitope of this mAb is located at a variable site. The structures of P1 and P40/P90 served as a topographic map to determine positions of known variable regions. RepMPs from P1 and P40/P90 are exclusively located at the N-terminal domains of these proteins, with the C-terminal domains remaining genetically conserved (Fig. 5b, c).

**Polyclonal antibodies against constructs of the C-terminal domain.** Adherence-inhibiting antibodies recognizing the constant C-terminal domain of P1 could be more effective in eliciting immunoprotection than antibodies to variable regions within the N-terminal domain. Previous studies had shown that gliding and binding of *M. pneumoniae* were affected by addition of a monoclonal antibody generated against a large recombinant fragment of P1 (residues Ala1160–Gln1518) that contains the C-terminal domain[12]. To consolidate these results and to better delineate the binding site of antibodies during an immune response, polyclonal antibodies were generated against two constructs of P1 spanning residues Lys1376–Asp1521 and Ala1400–Asp1521 (see "Methods"). The shortest construct corresponds accurately to the C-terminal domain, while the longest construct also includes the hinge between domains and the two strands from blade VII (Fig. 6a). Quantitative PCR (qPCR) was used to measure the amount of genomic DNA (gDNA) from the *M. pneumoniae* cells in solution, the fraction not adhered to the glass (see "Methods"). The *M. pneumoniae* cells had been previously incubated with different percentages of sera from rats challenged with the P1 constructs. The analysis determines how much the *M. pneumoniae* cells are affected by the presence of the polyclonal antibodies. Without exposure to sera, 81% of the *M. pneumoniae* cells attached to the plastic surface upon incubation of these (scraped) cells for 2 h in 24-well plates. Then, to evaluate the possible toxicity by other components of the sera, such as the complement that enhances the ability of antibodies and phagocytic cells to clear microbes, different percentages (1, 3, 6, and 10%) of serum obtained from pre-immune bleed (PPI) of challenged animals were tested. The total number of recovered mycoplasma after incubation only decreased with the highest concentrations (6 and 10%) of PPI when compared with samples not treated with serum (Supplementary Fig. 11a). Thus the low concentrations (1 and 3%) were selected to test the inhibitory effects on the adhesion capability of *M. pneumoniae* cells (Fig. 6b). Toxicity effects can be subtracted, at least in part, by normalization with the corresponding values of PPI serum and the measures for concentrations of 10% are also reported (Supplementary Fig. 11b). As a final checking, the specificity of antibodies for the C-terminal constructs was also assessed. Serum from challenged rats presented no significant differences with their PPI when incubated with the corresponding constructs before the quantitative adhesion assay.

Incubation of serum from rats challenged with the longest C-terminal domain construct resulted in the number of the *M. pneumoniae* cells attached being decreased significantly with reduction ratios with respect to PPI samples of 0.31 and 0.56 for added serum concentrations of 1% and 3%, respectively (*p* values of $9 \times 10^{-6}$ and $1 \times 10^{-6}$ by applying *t*-test one-tailed distribution; two samples equal variance). Reduction ratios for the short construct were 0.22 and 0.33, for added serum concentrations of 1% and 3%, respectively (*p* values of 0.016 and $3 \times 10^{-4}$). Together these results reveal that *M. pneumoniae* adhesion capabilities are significantly reduced by polyclonal antibodies generated against C-terminal domain constructs of P1, with inhibition being highest for the longest construct that includes the cytoadherence peptide.

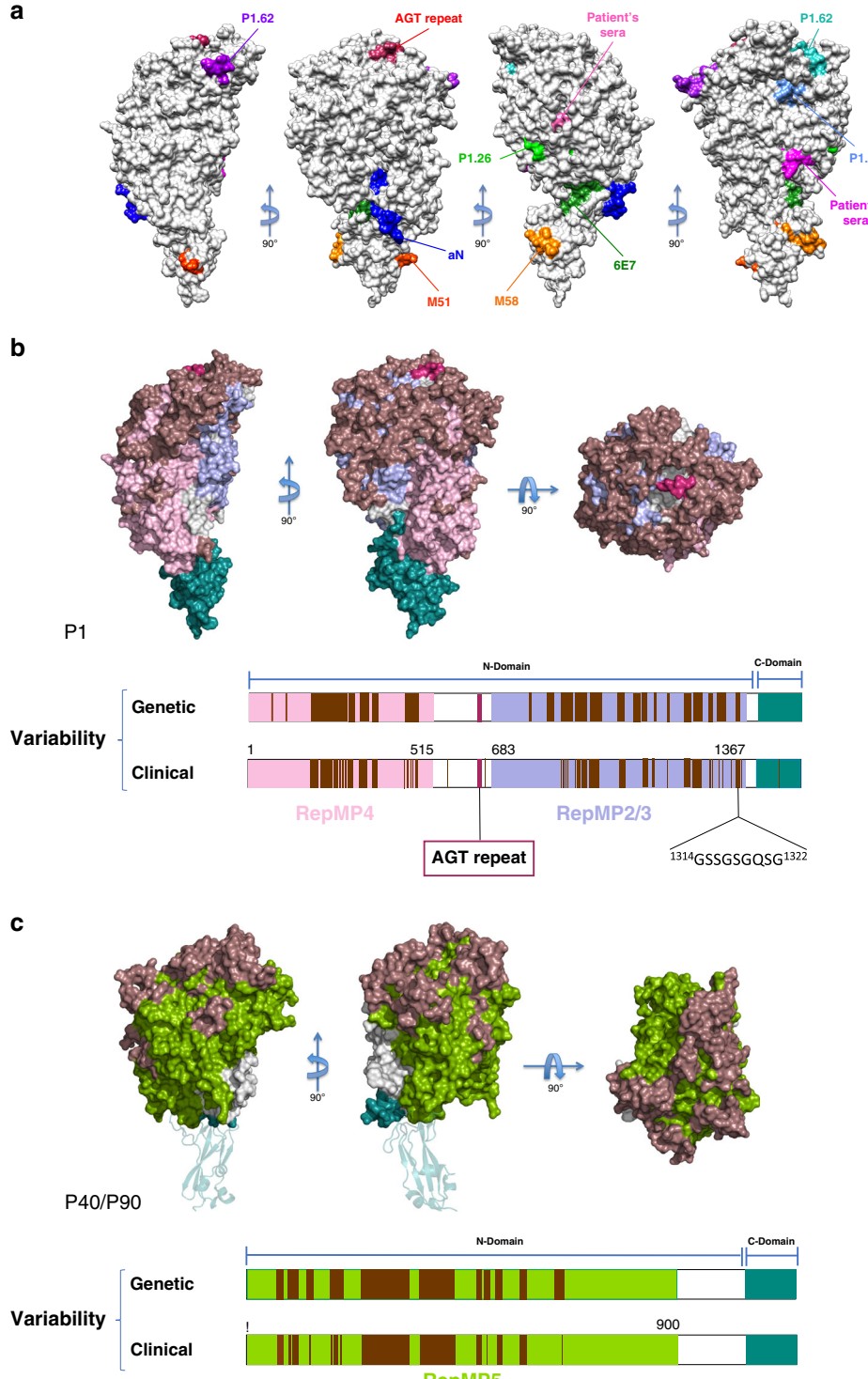

**Fig. 5 Mapping of P1 epitopes. Variability in P1 and P40/P90. a** Mapping into the P1 structure of all known epitopes (Supplementary Fig. 10). Most epitopes are located on the protein surface, although accessibility by antibodies appears to be difficult for a few of them. **b** Three views of the P1 surface, 90° apart from each other, with the genetically variable regions (brown), RepMP4 (light pink), RepMP2/3 (light blue), and the C-terminal domain (cyan). **c** The corresponding three views of the P40/P90 surface, with the genetically variable regions (brown), RepMP5 (green), and the C-terminal domain (cyan). The C-terminal domain, modeled from the structure of P110, is depicted as ribbons. Sites of genetic and clinical variability are also represented for P1 and P40/P90 in the lower part of the corresponding panels. Clinical variability refers to the variation exhibit by clinical isolates characterized so far. In turn, "genetic variability" indicates regions that are potentially variable by DNA recombination between RepMP elements in *M. pneumoniae* genome. Presently, genetic variability regions are wider than clinical variability.

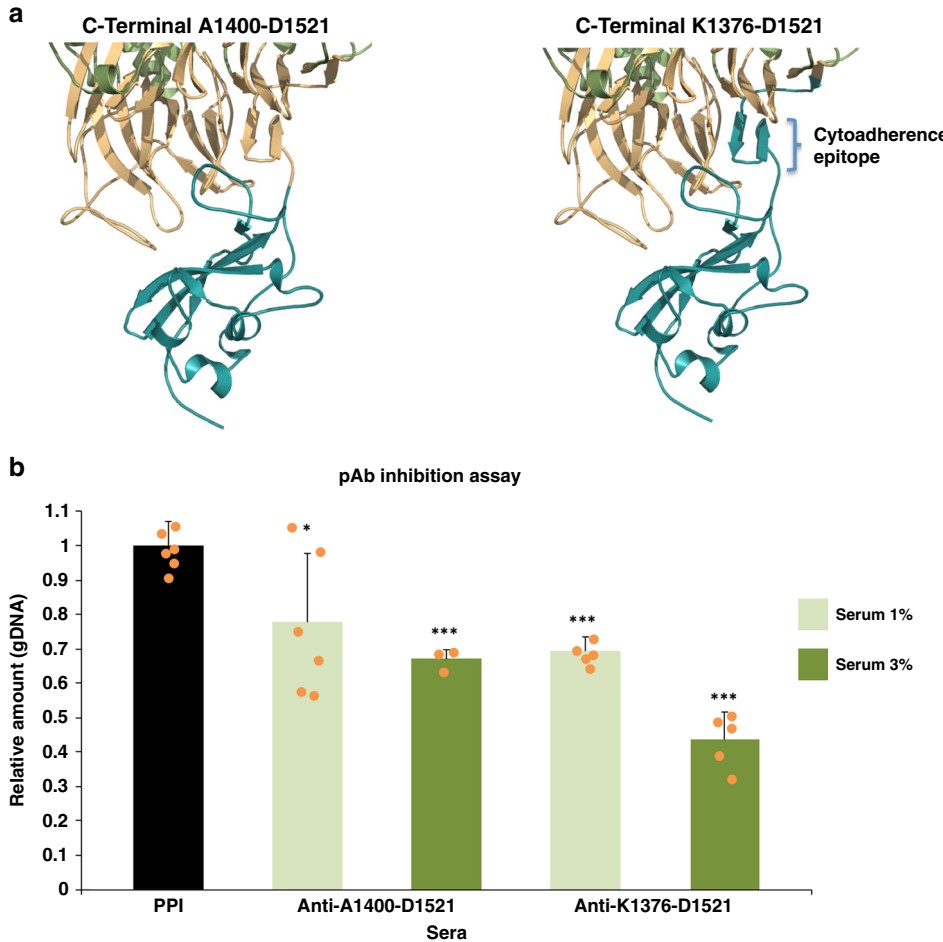

**Fig. 6 Inhibition adhesion assays of polyclonal antibodies generated against constructs of the P1 C-terminal domain. a** Residues corresponding to the P1 constructs used as antigens to generate polyclonal antibodies are colored in cyan in the structure of P1. The short construct (left, residues Ala1400–Asp1521) includes only the C-terminal domain. The long construct (right, residues Lys1376–Asp1521) includes also the hinge, which connects the N-terminal domain with the C-terminal domain, and the two strands from the last propeller β-sheet, also known as the cytoadherence epitope. **b** Sera raised against any of the two constructs inhibit adhesion of *M. pneumoniae* cells even at low concentrations, with 3% showing a significantly higher inhibition than 1%. Error bars represent the mean ± standard deviation. *$p < 0.05$ and ***$p < 0.001$. Source data are provided as a Source Data file.

**Recognition of P1 and P40/P90 by sera from *M. pneumoniae* patients**. Determination of the structural organization from the P1 and P40/P90 ectodomains encouraged us to assess the immunogenic activity of different constructs from these proteins using total sera from 19 patients with positive diagnostic for *M. pneumoniae* based on serological tests (Table 1). Clinically relevant information, including the IgG and IgM values, was available for all sera. In addition, sera from five patients that tested negative for *M. pneumoniae* were also included in the study for control purposes. First, the immunodominant nature of the adhesins was confirmed, revealing that 17 out of the 19 sera (90%) tested positive for P1. Similarly, 14 sera (74%) tested positive for P40/P90; all positive sera for P40/P90 were positive for P1. Of note, eight sera (42%) were also positive for P140. This result is not unexpected because P1 and P140 show extensive antibody cross-reactivity[43]. Alternatively, some of the P140 positive patients might have been infected with *M. genitalium* in addition to *M. pneumoniae*. Next, the sera response was examined with the same two C-terminal domain constructs of P1 (Lys1376–Asp1521 and Ala1400–Asp1521) described before in the generation of polyclonal antibodies. The number of positive sera, 5 out of 19 (26%), decreased substantially with respect to the full length P1, although importantly all these positive sera correspond to the ones with highest IgG levels.

For P40/P90, an additional evaluation of the sera reactivity with recombinant proteins lacking either insertion 1 (S1), or insertions 1 and 2 (S1S2) was performed to assess the contribution of the insertions and the loss of the proteolytic site. Construct S1 was detected by 14 out of 19 sera (74%), the same ratio to that of the entire P40/P90. In contrast, remarkably, there is an important decrease to just seven positive sera (37%) when the P40/P90 S1S2 construct was evaluated. Furthermore, even for most of these seven positive sera the response was substantially reduced. Therefore, results indicate that P40/P90 is a strong immunogenic protein, with insertion 2 (S2) acting as an immunodominant region, although recognized by sera with low IgG levels.

## Discussion

Adherence of *M. pneumoniae* to sialylated receptors on the surface of respiratory cells was elucidated years ago[34,35,37,44]. Since the identification of the critical role of P1 in *M. pneumoniae* cytoadherence, it has been accepted that this protein was responsible for binding to sialic acid oligosaccharides[11–14,45]. In addition to P1, other cytadherence-accessory proteins had been reported as required for binding to sialic acids[14,46]. The present

**Table 1 ELISA immunoassays for antigen detection by sera from patients infected with *M. pneumoniae*.**

| Antigen | 1 | 2 | 3 | 4 | 5 | 6 | 7 | 8 | 9 | 10 | 11 | 12 | 13 | 14 | 15 | 16 | 17 | 18 | 19 |
|---|---|---|---|---|---|---|---|---|---|---|---|---|---|---|---|---|---|---|---|
| P1 | ++ | ++ | ++ | ++ | ++ | ++ | ++ | ++ | ++ | ++ | +++ | ++ | + | ++ | ++ | − | +++ | − | +++ |
| C-terminal 1376 | − | − | ++ | − | − | ++ | − | − | − | − | +++ | − | − | − | − | − | +++ | − | +++ |
| C-terminal 1400 | − | − | ++ | − | − | − | − | − | − | − | +++ | − | − | − | − | − | + | − | +++ |
| P40/P90 | ++ | ++ | − | ++ | ++ | ++ | ++ | ++ | ++ | − | +++ | − | ++ | +++ | +++ | − | +++ | − | +++ |
| P40/P90s1 | ++ | ++ | − | ++ | ++ | ++ | ++ | ++ | +++ | − | +++ | − | ++ | +++ | +++ | − | +++ | − | +++ |
| P40/P90s1s2 | − | − | ++ | + | − | − | − | − | − | − | ++ | − | − | − | ++ | + | ++ | − | ++ |
| P140 | − | − | − | − | − | − | − | − | − | − | − | − | − | − | − | − | − | − | − |
| **IgG–IgM levels**[a] | | | | | | | | | | | | | | | | | | | |
| IgG (UA/ml) | 42.8 | 35.7 | 62.6 | 4.65 | 0.44 | 0.72 | 34.1 | 0.25 | 2.15 | 14.3 | >200 | 15.9 | 33.6 | 64.4 | 49.50 | 25.2 | >200 | 44.5 | 72.4 |
| IgM (UA/ml) | >27 | 2.8 | 17 | 14 | 15 | >27 | >27 | 12 | >27 | 19 | 2.4 | >27 | >27 | >27 | 12.00 | 2.80 | 6.20 | 1.80 | 7.40 |

−, negative (value < 0.080); +, weak positive (0.080 ≤ value ≤ 0.100); ++, strong positive (value > 0.100).
[a]IgG and IgM levels against a P1 C-terminal fragment of *M. pneumoniae* determined by the commercial Liaison *M. pneumoniae* IgG, IgM kit (DiaSorin).

study demonstrates that the binding in vitro to sialic acid receptors relies in fact on P40/P90 (Fig. 3c and Supplementary Fig. 7), however, affinities and specificities could be tuned through other proteins or due to the density and distribution of the oligosaccharides in the surface[37,38]. This important result is consistent with the recent finding that binding to sialic acid oligosaccharides is mediated by P110 in *M. genitalium*[18]. Cleavage of the P40/P90 polypeptide chain occurs in vitro at two exposed Arg–Ala motives (Arg445–Ala446 and Arg455–Ala456) located in the central part of the N-terminal domain, producing two polypeptide chains that can correspond to the P40 and P90 subunits found in vivo (Supplementary Figs. 4 and 5a). Cleavage appears to be a divalent metal ion dependent auto-proteolysis taking place only after the folding is completed. Nevertheless, P40/P90 continues to behave as a single globular protein structurally similar to P110. This is in agreement with the extensive interface between the P40 and P90 polypeptides indicating that they form a very stable ensemble (according to PISA[32]) (Supplementary Fig. 6). However, P40 is located at the outer surface part of the Nap and due to the absence of covalent bonds might become separated from P90 and from the Nap complex under stringent or denaturing conditions. The loss of P40 only in certain conditions could explain the discrepancies in the past about whether or not P40 was part of the adhesion complex. The sialic acid oligosaccharides binding site is located in a loop of the P90 polypeptide with P40 contributing only to the entrance of the pocket (Fig. 3c and Supplementary Fig. 6).

There are clear structural relationships between P1 and P40/P90 suggesting a common protein ancestor, a hypothesis that is reinforced by the structural similarities found between P1 with the non-orthologous protein P110 from *M. genitalium* (Fig. 4 and Supplementary Fig. 9). This common ancestor could have been encoded in a mobile element, thus facilitating transmission within different species of the pneumoniae cluster. The presence of multiple copies of the adhesin genes in the genomes of several mycoplasmas supports this hypothesis. In each species, the adhesin genes may have evolved to mediate adherence to different hosts and tissues. Molecular systems where one of the catalytic sites, at the interface of two paralogs, is lost during evolution can be related with the situation in F-type ATPase/synthase, widely distributed among most living organisms[47], raising the possibility of a similar evolutionary process for the Nap. Interestingly, the sequence identity between P40/P90 and P1 is low in the N-terminal domains (12%), increases in the C-terminal domains (17%), and becomes high for the transmembrane and cytoplasmic regions (43%). The increased variation of the sequences from the most exposed regions of the Nap, the N-terminal domains, can reflect specific attachment and motility requirements but also a mycoplasma survival strategy related with enhanced antigenic variability.

To date, 13 antigenic variants have been identified in clinical isolates for P1 and 5 for P40/P90[48] (Fig. 5b, c). Regions of P1 and P40/P90 presenting genetic and clinical variability cluster at the surface of the upper part of the N-terminal domain, where antibodies could easily block access to the receptor binding site. In fact, mapping the known genetic variability of *M. pneumoniae* onto the Nap structure, modeled according to the information available from *M. genitalium* with the structures of P1 and P40/P90, reveals that the sialic binding site pocket is completely surrounded by these variable regions (Fig. 7). P1 and P40/P90 present many sequence insertions, which amount to more than 200 extra residues in each protein, with respect to their orthologues P140 and P110 in *M. genitalium* (Supplementary Figs. 1 and 4). Interestingly, most of the insertions correspond to disordered regions in the structures of the *M. pneumoniae* proteins. In P1 there are 14 disordered loops, while in P40/P90 there are 4,

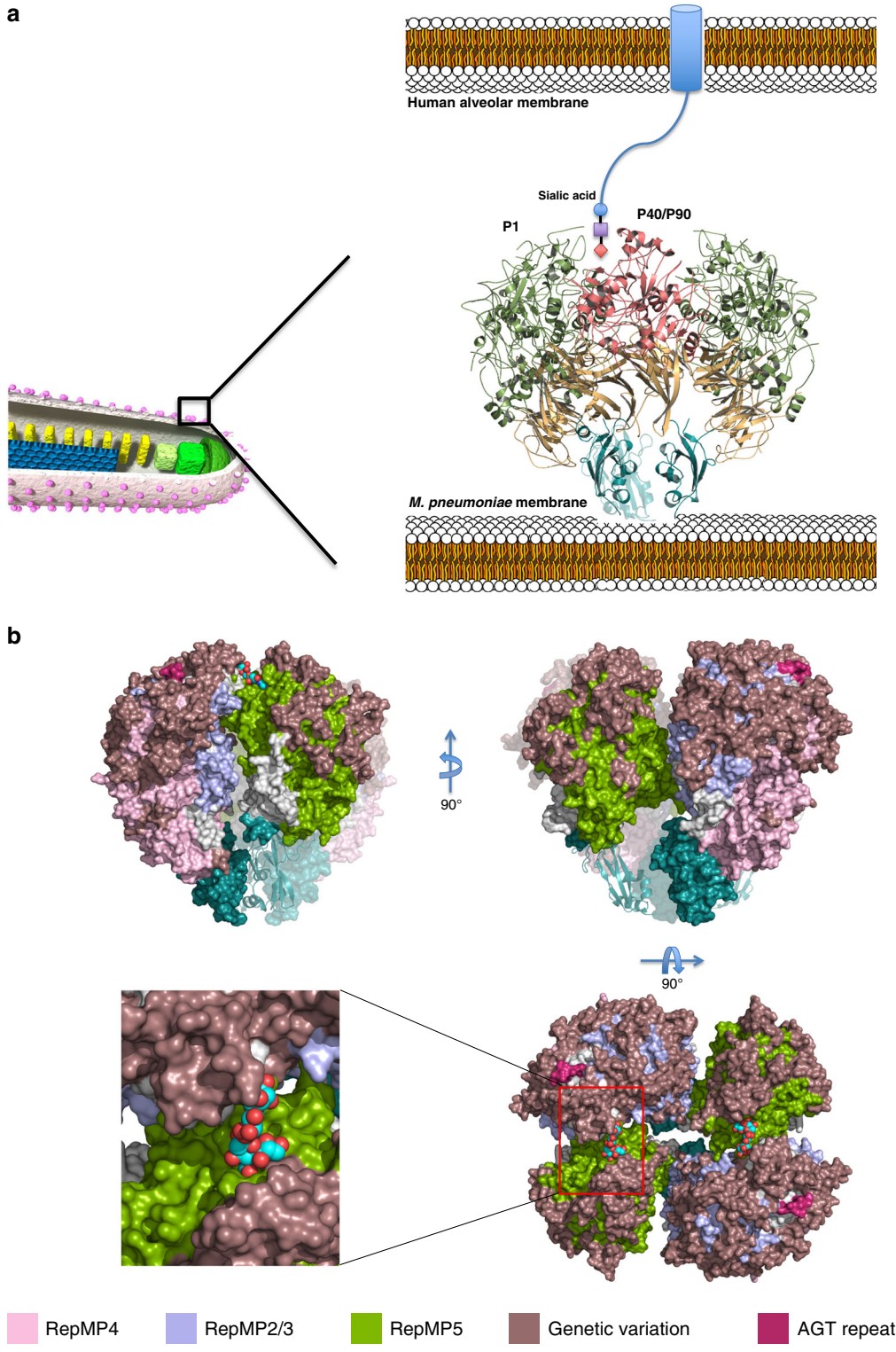

| RepMP4 | RepMP2/3 | RepMP5 | Genetic variation | AGT repeat |

**Fig. 7 Structure of the *M. pneumoniae* Nap. a** Representation of the whole *M. pneumoniae* Nap ectodomain as a dimer of a P1-P40/P90 heterodimer, modeled from the information recently available for *M. genitalium*. Binding to a receptor from the host cell, a human alveolar epithelial cell, is indicated. The sialic acid binding site is located in a pocket at the top of the Nap that is not far from the twofold symmetry axis of the Nap. The crowns from P1 (green) and P40/P90 (pink) are separated by the β-propellers (brown) from the C-terminal domains (cyan), which anchor the Nap to the mycoplasma membrane. **b** Three views, 90° apart from each other, of the Nap surface depicting the sialic acid compound 3SL (spheres) bound into the binding site of the two P40/P90 subunits of a Nap. Regions with genetic variability of P1 and P40/P90 (brown) cover most of the surface from the upper part of the Nap. RepMP5 from P40P90 is depicted in green, while RepMP4 and RepMP2/3 from P1 are depicted in pink and blue, respectively. C-terminal domains from both P1 and P40/P90 are depicted in cyan. The inset shows the sialic binding site pocket, with a 3SL molecule bound, surrounded by genetically variable regions.

but 2 of them more than 60 residues long (insertions S1 and S2). The C-terminal domain from P1 contains only one disordered loop and no genetic variability, pointing to the interest of immunogenic studies with this domain.

Polyclonal antibodies against two constructs from the C-terminal domain of P1 (Lys1376–Asp1521 and Ala1400–Asp1521) inhibit the cytoadherence of *M. pneumoniae* cells (Fig. 6b). In addition, most of the sera from *M. pneumoniae* infected patients presenting a high IgG content also recognize these C-terminal domain constructs (Table 1). On the contrary, these C-terminal domain constructs were poorly recognized by sera from the early stages of infection where a high IgM content is present. The full length extracellular region of P1 was strongly recognized by all sera, independently of the IgG or IgM ratios, confirming the immunodominance of P1[19]. These results indicate that the C-terminal domain might be a relatively weak or poorly accessible antigenic region, in agreement with what the structural information suggests, but effective to generate more specific IgGs antibodies capable of inhibiting mycoplasma cytoadherence. Large P1 constructs, such as the whole extracellular region[49] or a long commercial peptide (residues 1160–1521, LIAISON *M. pneumoniae* IgM and IgG, Biotrin International Ltd), maybe better suited for diagnosis than small C-terminal domain constructs, which might, however, provide a significant protective response. P40/P90, often overlooked in immunological studies in spite of pioneering studies showing P90 was an important immunogen[50], is recognized by 74% of the sera from infected patients. Interestingly, the serologic response decreased sharply for P40/P90 constructs when the second insertion (S2 residues Leu395–Thr462) was eliminated. Most of the sera that recognize S2 have a high IgM content, dominant in the initial stages of the immune response. S2 presents the highest genomic (64%) and clinical (~80%) variability and is highly exposed and mobile on the surface of P40/P90. Exposure of S2 is likely increased by the cleavage at its C-end that generates the two polypeptides P40 and P90. Altogether these results suggest that S2, located within P40, might be acting as a decoy to divert the immune response (Table 1). The high immune response against P40/P90, with unique structural peculiarities, invites a reconsideration of its therapeutic possibilities.

Gliding of mycoplasmas from the pneumoniae cluster, including *M. pneumoniae* and *M. genitalium*, is thought to be caused by a mechanism where the force, generated by ATP energy[51], is transmitted to the Nap that appears to work as a leg/foot on the surface of the mycoplasma cell[7,8,12,52–56]. Gliding of *M. pneumoniae* cells is always in the direction of the attachment organelle, which indicates that the gliding machinery, including the Naps at the cell surface, should provide a well-defined directionality. To achieve a directional gliding, the release of the sialylated receptors is as essential as the binding because mycoplasma cells cannot move forward if they remain anchored to the surface receptors where they have first attached. In the Nap, the conformational changes required by the foot/leg activity during gliding have to be synchronized with the binding/release to the cell receptors, which are thought to be randomly distributed on the host surface. The binding site in P40/P90 could be regulated by the interaction with P1 as found in *M. genitalium*[17]. This may be the reason why P1 was expected to contain the binding site. In the cryo-EM map of P1, the weak density of the C-terminal domain indicates flexibility with respect to the N-terminal domain proving that P1 can experience significant conformational changes consistent with the observations that antibodies that bind to the C-terminal domain can interfere with the attachment and gliding of mycoplasma cells. Therefore, although P40/P90 contains the binding site to the sialic acid cell receptors,

P1 has still to be seen as a major player in the Nap functioning during gliding and also in the adhesion to host cells. Moreover, proteins others than the Nap components, including P30, P65, HMW1, HMW2, or PrpC/PrkC, are known to be essential for binding and gliding[57–60]. These proteins might support functioning and conformation of the Naps to play its roles.

Integration of previous knowledge with the structural data obtained in this work provides important clues for a better understanding of the functioning of Naps in *M. pneumoniae*, although many questions remain open about the adhesion and gliding motility mechanisms. The structural framework obtained in this work also allows the mapping of epitopes and of the genetic and clinical variability of P1 and P40/P90 explaining many of the immunogenic properties of these immunodominant proteins in *M. pneumoniae*. These results can help in the development of vaccines and therapeutics for inhibiting the infectivity of this respiratory pathogen.

## Methods

**Cloning, expression, and purification of P1 and P40/P90 constructs**. Regions corresponding to the MPN141 and MPN142 genes from *M. pneumoniae* were amplified from synthetic clones (Supplementary Tables 3 and 4), using primers P1F and P1R and P40/P90F and P40/P90R, respectively (Supplementary Table 5). The PCR fragments were cloned into the expression vector pOPINE[61] (gift from Ray Owens plasmid #26043, Addgene, Watertown, USA) to generate constructs, with a C-terminal His-tag, comprising residues 29–1521 for P1 and 23–1003 and 23–1114 for a short and a long construct of P40/P90, respectively. Recombinant proteins were obtained after expression at 22 °C in B834 (DE3) cells (Merck), upon induction with 0.8 mM IPTG at 0.6 OD$_{600}$. Cells were harvested and lysed by sonication in 1xPBS, 40 mM imidazole and centrifuged at 49,000 × g at 4 °C. Supernatant was loaded into a HisTrap 5 ml column (GE Healthcare) pre-equilibrated in 1xPBS with 40 mM imidazole as binding buffer and 1xPBS with 400 mM imidazole as elution buffer. Soluble aliquots were pooled and loaded onto a Superdex 200 GL 10/300 column (GE Healthcare) in buffer consisting of TRIS 20 mM pH 7.4 and 150 mM NaCl. Buffers prepared for P40/P90 purification were supplemented with 1 mM EDTA to prevent proteolysis. Mutants of P40/P90 were produced following the same protocol as for the wild type, but using the corresponding primers showed in Supplementary Table 5. Cleavage sites in P40/P90 were determined with N-terminal Edman sequencing of the bands obtained by SDS-PAGE of purified samples of the P40/P90 constructs (Supplementary Fig. 5a). Mixtures of P1 with either the short or the long P40/P90 constructs were used for the preparation of complexes in a 1:1 ratio. The presence of stable P1-P40/P90 heterodimers, verified by gel filtration, was the only complex detected in the mixtures (Supplementary Fig. 5c).

For cryo-EM data collection, P1 of *M. pneumoniae* M129 strain (MPN141) was expressed and purified as described previously[62]. The *p1* gene corresponding to the amino acid residues 60–1518 of P1 adhesin was codon-optimized, synthesized, and inserted between *Nde*I and *Xba*I sites in pCold I vector (TAKARA BIO, Shiga, JAPAN). The resulting plasmid, pP1-1, wherein *p1* was fused to an N-terminal 6 × His-tag and a Factor Xa site. BL21 or BL21 (DE3) pLysS cells carrying pP1-1 were cultured at 37 °C until the culture density reached an OD$_{600}$ value of 0.3. The cells were subjected to cold shock at 15 °C for 30 min, followed by the addition of IPTG to a final concentration of 1.0 mM, and cultured overnight at 15 °C. The cells were harvested by centrifugation (4000 × g, 10 min, 4 °C) and washed twice with a TN buffer comprising 20 mM Tris-HCl and 500 mM NaCl. The suspension was frozen at −80 °C. The cell pellet was thawed, suspended in a binding buffer (20 mM Tris-HCl pH 8.0, 500 mM NaCl, 20 mM imidazole) containing 1 mM PMSF. Cells were then disrupted with a Sonication (NIHONSEIKI, Tokyo, Japan). Unbroken cells were removed by centrifugation (60,000–140,000 × g, 60 min, 4 °C). The supernatant was loaded onto a HisTrap HP column (GE Healthcare, Milwaukee, WI) equilibrated with the binding buffer. The bound proteins were eluted with a linear gradient of 10–200 mM imidazole in TN buffer. Fractions containing P1 adhesin were dialyzed against a TNG buffer (20 mM Tris-HCl (pH 8.0), 150 mM NaCl) and loaded onto a HiLoad 16/60 Superdex 200 prep grade column (GE Healthcare) equilibrated with the TNG buffer at a flow speed of 1 mL/min. The collected P1 adhesin fractions were combined, dialyzed against 20 mM Tris-HCl (pH 8.0), and applied to an anion exchanger, Q Sepharose Fast Flow equilibrated with the buffer, and eluted with a linear gradient of NaCl from 0 to 0.4 M. The purified protein was concentrated using an Amicon Ultra 30k spin filter (Millipore, Darmstadt, Germany) if necessary.

**Crystallization and X-ray data collection of P1 and P40/P90N**. Ninety-six-well plates crystallization screenings were performed on both Cartesian (Cartesian (TM) Dispensing Systems) and Phenix (Art Robbins Instruments) robots mixing 150 nL of reservoir condition with 150 nL of protein solution at concentrations of 6 mg/mL for each of the P1 and P40/P90 proteins. The optimized crystals of P1 were

obtained after mixing 1 μL of P1 at 6 mg/mL with 1 μL 20% PEGMME 2000 and PCB (sodium propionate, sodium cacodylate, and BIS-TRIS propane pH 4) as a reservoir condition in a ratio of 2:1, respectively. P40/P90N optimized crystals were obtained mixing 1 μL of P40/P90N at 6 mg/mL with 1 μL 20% PEG 3350, 0.3% MetOH in a ratio 1:1, respectively. To obtain crystals of P40/P90N in complex with 3SL or 6SL, protein samples were previously incubated with 10 mM of each oligosaccharide 1 h at 20 °C. All attempts for P40/P90 crystallization were not successful. Crystals used for data collection were flash frozen in liquid nitrogen with 15% glycerol as cryo-protectant.

X-ray data collection was carried out at Xaloc beamline (ALBA Synchrotron, Spain). Data were processed with Xia2[63] using XDS[64], Aimless and Pointless[65] from the CCP4 suit of programs[66].

**Determination of the P1 crystal structure**. The crystals from P1 belong to space group C2 and could contain, by packing considerations, only one subunit in the asymmetric unit. We tried to solve these P1 crystals by heavy atoms and SeMet derivatives, but finally the resolution was achieved by density modification techniques applying solvent flattening and averaging with crystals from the *M. genitalium* orthologous protein P140, whose structure was also unknown at that time[17]. With this procedure the structures of P1 and of P140 (with a sequence identity of 41% between their ectodomains) were solved simultaneously. The P140 containing crystals used corresponded to the complex of P140 with the N-domain of P110 (P140–P110N) and to crystals from P140 alone. Steps followed for the structure determination of P1 and P140 were the following:

(1) A partial molecular replacement solution was obtained for the P140–P110N crystals using as searching model the N-domain from the P110 structure[17]. This partial solution of the P140–P110N crystals contained four P110N subunits in the asymmetric unit. The four P110N subunits were organized as two pairs and within each pair the two subunits were related by accurate local (non-crystallographic) twofold axis. In this partial solution, no density was visible that could correspond to P140.

(2) From this partial solution of the P140–P110N crystals, a difference (Fo–Fc) map was generated at about 3 Å resolution. Positive densities in this difference map, expected to correspond mainly to P140, were tentatively reinforced by averaging (without applying any mask) using one of the local twofold symmetries. This unmasked averaging also weakens density from other subunits in the crystal not related by the local twofold symmetry. In this averaged density it was possible to differentiate continuous regions of strong and of weak density, but it was not yet possible to identify any consisting molecular feature from P140.

(3) The cryo-electron tomography (CET) map of a whole *M. genitalium* Nap[17] was then used to define an initial possible mask for P140. The CET map, at ~17 Å resolution, corresponded to a dimer of P140–P110 complexes, where the fitting of the P110 structures was unambiguous. The CET map was placed on the (Fo–Fc) averaged map by superposing the corresponding fitted P110 structures. Then, a putative "P140 initial map" was generated as the (Fo–Fc) averaged density inside the CET envelop corresponding to P140. After some polishing (avoiding overlapping with neighbor subunits or retaining regions with continuous densities) the assumed "P140 initial map" was used to perform a molecular replacement search (using program PHASER) obtaining a reasonable possible solution with four P110N structures and four "P140 initial maps." Attempts to use the density from the CET maps directly for phasing, and not just as a mask, were unsuccessful in our hands.

(4) Iterative cycles of density averaging (and solvent flattening), alternating with manual readjustments of the P140 mask, were then performed with the P140–P110N crystals. The procedure converged giving clearer and more continuous density, although no secondary structures that could correspond to P140 were visible. Likely, the almost parallel orientation of the two non-crystallographic twofold axes weakened the phasing power of this averaging. In spite of the limitations, the new density (within the updated mask) allowed obtaining a molecular replacement solution for the P1 crystals. Surprisingly, the new density did not provide a solution for the P140 alone crystals.

(5) Density modification with averaging within the P140–P110N crystal and now also with the P1 crystals (using program DMMULTI[67]) improved quickly the maps. The iterative cycles of DMMULTI were alternated with cycles of phase extension for the P1 crystals (using the "autobuild" protocol in Phenix at 1.94 Å resolution) and with manual readjustments of the averaging masks (that were updated about forty times). Model building was carried forward for P1 and P140 in parallel. With about 50% of the structures traced, it was possible to obtain a molecular replacement solution for the P140 alone crystals that, with six subunits in the crystal asymmetric unit, facilitated the completion of the P1 and P140 structures.

P1 was then refined, giving agreement factors $R$ and $R_{free}$ of 18.7% and 22.9%, respectively (Supplementary Table 1). The difference between these two factors was used traditionally to assess how well the available molecular model explains the corresponding experimental diffraction data. The difference between both factors is related with the bias introduced in the model during refinement. Despite the

quality of the final map (Fig. 2b), the P1 structure presented a significant amount of disordered residues located at the N-end (missing residues 29–59) and in 14 loops (102–105, 228–230, 259–268, 278–282, 298–300, 337–348, 831–847, 870–888, 923–928, 941–944, 1226–1232, 1308–1324, 1341–1349, and 1482–1495). The final refined structure has been deposited in the PDB with code 6RC9.

**Determination of the P40/P90 structures**. The P40/P90 crystals belong to space group $P2_12_12_1$ and contain two subunits in the asymmetric unit. The structure was solved by molecular replacement with program Phaser[68] using as searching model the structure from the orthologous protein P110 from *M. genitalium* (PDB code 6R3T). The structure of P40/P90 was refined at 2.65 Å resolution, giving agreement factors $R$ and $R_{free}$ of 21.4% and 23.4%, respectively (Supplementary Table 1). Crystals of complexes from P40/P90 with 3SL and 6SL belong to space C2, although with pseudo-orthorhombic unit cell dimensions, and contain one subunit in the asymmetric unit. The 3SL crystals showed a well-defined oligosaccharide with high occupancy, at 3.2 Å. The 6SL crystals, at 2.8 Å resolution, presented some disorder and an overall weak density for the oligosaccharide that was interpreted as partial occupancy.

Tracing and refinement of all the structures was performed alternating interactive and automatic cycles with programs Coot[69] and Buster[70], respectively. The final refined structures have been deposited in the PDB with codes: 6RJ1 for P40/P90, 6TLZ for the 3SL complex, and 6TM0 for the 6SL complex.

**Single-particle cryo-EM**. The purified P1 protein construct was applied onto a Quantifoil holey carbon grid (R1.2/1.3, Cu, 200 mesh) covered with a thin film of GO flakes (SIGMA-ALDRICH), blotted for 4.5 s at 4 °C in 100% humidity, and plunge frozen in liquid ethane by using a Vitrobot Mark IV (Themo Fisher Scientific). Cryo-EM imaging was obtained using a Titan Krios (Themo Fisher Scientific) operating at 300 kV acceleration voltage and equipped with a Cs corrector (CEOS, GmbH) and a Falcon III direct electron detector (Themo Fisher Scientific). A total of 5279 movies were obtained in the electron counting mode with a physical pixel size of 0.69 Å/pixel and total dose of 60 e/Å$^2$ with 36.33 s exposure time. The data were automatically collected using EPU software with a defocus range of −0.75 to −2.75 μm and were fractionated into 78 movie frames.

**Single-particle cryo-EM image processing**. The movie frames were subsequently aligned to correct for beam-induced movement and drift using MotionCor2[71], and contrast transfer function was estimated using Gctf[72]. A total of 4,574,424 particles images were automatically picked using Gautomatch (http://www.mrc-lmb.cam.ac.uk/kzhang/) and several rounds of 2D classification and 3D classification were performed using RELION-2.1[73]. A total of 295,195 particle images from good 2D classes were selected for making the initial model of P1 using cryoSPARC[74]. First and second 3D classification were calculated with angular sampling interval at 7.5 and 3.7 degrees, respectively. Finally, 68,014 particles from the best 3D class were 3D refined, producing a reconstruction with a resolution of 2.88 Å and a B-factor of 89 Å$^2$ with the gold-standard FSC criteria (FSC ≥ 0.143). The local resolution was estimated using RELION-2.1. The processing strategy is described in Supplementary Fig. 2a–d and the model refinement statistics in Supplementary Table 2.

**SEC–MALS analysis**. The molecular weight of the P40/P90 short construct was measured using a Superose 6 10/300 GL (GE HEalthcare) column in a Prominence liquid chromatography system (Shimadzu) connected to a DAWN HELEOS II multi-angle light scattering (MALS) detector and an Optilab T-REX refractive index (dRI) detector (Wyatt Technology). ASTRA 7 software (Wyatt Technology) was used for data processing and result analysis. A dn/dc value of 0.185 ml/g (typical of proteins) was assumed for calculations.

**Antibody production and validation**. Polyclonal antibodies against adhesin P1 were produced by Eurogentec. Briefly, rats were injected with peptides Lys1376–Asp1521 or Ala1400–Asp1521 and bleeds were collected at day 28. Pre-immunization bleeds were also collected as negative controls. Specificity of the polyclonal antibodies was tested by western blot against the recombinant peptides and protein extract of Mycoplasma cells. The dilution used for each antibody was 1:500.

The production of polyclonal antibodies has complied with ethical regulations for animal testing and research. It complies with the following association's requirements: Federation of European Laboratory animal Science Associations and UK Home Office Animals Scientific Procedures Act.

The regulations followed by Eurogentec are: 2010/63/EU and 01/2005/EU.

**Quantitative adhesion assay**. *M. pneumoniae* (ATCC 29343) was grown in a T75 flask in Hayflick medium during 24 h, harvested by scraping, and resuspended in Hayflick medium to a final concentration of $10^7$ cells/ml. Incubation, during 1 h at room temperature on a rotating wheel, of 1 ml of a cell suspension with sera (at 1, 3, 6, and 10% concentrations) containing polyclonal antibodies raised against peptides Lys1376–Asp1521 or Ala1400–Asp1521, in presence or absence of 2 μg from the corresponding peptide. A preimmunization serum was used as a negative control. After incubation, cell suspensions were seeded in 24-well plates and

incubated at 37 °C to allow cells adhere to the plastic. After 2 h, supernatants (containing the fraction of the cells that did not adhere to the plastic) and attached cells were collected and processed for gDNA extraction using MasterPure DNA Purification Kit (EpiCentre #MCD85201). Samples were obtained in triplicate, and the relative amount of cells was quantified by qPCR with SYBR Green PCR Master Reagent (Thermo #4367659), using a LightCycler 480 Real-Time PCR machine (Roche) with the following conditions: denaturation at 95 °C for 10 s; 40 cycles of amplification of 95 °C for 15 s and 60 °C for 1 min; melting curve at 95° for 15 s, 60 °C for 15 s, 95° continuous. The following oligonucleotides were used: qPCR Myco FW 5′-ACGATGATTACAGGCGGTTC-3′ and qPCR Myco RV 5′- GTTG GTGGCCTCTTGTTGAT-3′.

**Immunoassays.** Diagnostic of *M. pneumoniae* was conducted using the Liaison *M. pneumoniae* IgG, IgM kit (DiaSorin) using a 1/100 dilution of the patient sera. Indirect ELISA assays were performed on 96-well plates Immulon 4 HBX 96-well plates (ThermoFisher) incubating 1 μg of each antigen at 4 °C overnight. 1/100 dilutions of each patient serum were added to the plate and detected using an anti-human IgG antibody conjugated with HRP (Thermofisher Scientific). Upon incubation for 30 min with 100 μl of substrate (Thermofisher Scientific), 100 μl of sulfuric acid 25% was added to stop the reaction and absorbance was read at 450 nm on a Triturus ELISA instrument (Grifols) device. Reference filter was set at 620 nm.

**Ethical approval.** Ethical approval for the study was obtained from the Ethics Committees for Research with Drugs from the Parc Taulí (Ref 2019/664) and Vall d'Hebron University Hospitals (PR(AG)24/2020).

**Inductive coupled plasma–mass spectrometry (ICP–MS).** Samples were digested with 1 ml of $HNO_3$ plus 0.5 ml of $H_2O_2$ in the closed teflon reactor during 24 h at 90 °C. The Zn measure was realized with ICP–MS in an instrument Agilent 7500ce with respect to a calibration line prepared by dilution from a ZN standard solution (Inorganic Ventures).

**Surface plasmon resonance.** Binding kinetics were determined by SPR using a Biacore 3000 biosensor platform (GE Biosystems) equipped with a research-grade streptavidin-coated biosensor chip SA. The chip was preconditioned with three 1-min injections of 1 M NaCl and 50 mM NaOH. Subsequently, the second and third flow cells of the chip were loaded, respectively, with oligosaccharides 6SL-PAA-biotin and 3SL-PAA-biotin (Carbosynth) at 10 μg/ml diluted in 20 mM TRIS pH 7.4, 150 mM NaCl, and 0.05 % tween20. It is noteworthy to mention that the ligands used contains a long polyacrylamide chain connecting the biotinylated chemically linked tag with the oligosaccharides, which should avoid most steric clashes between biotin and the P1 and P40/P90 samples used. The immobilization levels acquired were at ~200 response units and the first cell was left blank to serve as a reference. A series of diluted concentrations were injected using a flow rate of 30 μl/min at 25 °C (0.9, 1.9, 3.7, and 7.5 μM of P40/P90 for 6SL titration and 1.0, 2.0, 4.0, 16.0, and 32.0 μM of both P40/P90 and P1 for 3SL titration). The proteins were allowed to associate and dissociate for 60 and 120 s, respectively, for P1 and 120 and 240 s, respectively, for P40/P90. Dissociations were followed by a regeneration step of 30 s with 0.05% SDS at 30 μl/min. The data were fitted assuming a Langmuir 1:1 binding model using BiaEvaluation 3.1 software to determine the equilibrium dissociation constant $K_D$.

**Reporting summary.** Further information on research design is available in the Nature Research Reporting Summary linked to this article.

## Data availability

Atomic coordinates and structure factors for the reported crystal structures of P1, P40/P90, and the P40/P90-3SL and P40/P90-6SL complexes have been deposited into the Protein Data Bank (PDB) under accession codes 6RC9, 6RJ1, 6TLZ, and 6TM0, respectively. Cryo-EM density has been deposited in the EM Data Base under the accession code EMD-30233. Atomic coordinates of the Cryo-EM structure have been deposited in the PDB under the accession code 7BWM. Source data are provided with this paper. Other data are available from the corresponding authors upon reasonable request.

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

## Acknowledgements

This work was supported by grants BFU2018-101265-B-100 (MINECO) to I.F., BIO2017-84166-R (MINECO) to J.P., and BES-2015-076104 (MINECO) to M.L.S., JSPS KAKENHI Grant Number JP25000013 to K.N., and a FEDER project from Instituto de Salud Carlos III (ISCIII, Acción Estratégica en Salud 2016). This work has also been funded by the Platform Project for Supporting Drug Discovery and Life Science Research (BINDS) from AMED under Grant Number JP19am0101117 to K.N. (support number 1282), by the Cyclic Innovation for Clinical Empowerment (CiCLE) from AMED under Grant Number JP17pc0101020 to K.N., and by JEOL YOKOGUSHI Research Alliance Laboratories of Osaka University to K.N. This work was supported by a Grant-in-Aid for Scientific Research on the Innovative Area "Harmonized Supramolecular Motility Machinery and Its Diversity" (MEXT KAKENH, JP24117002 to M.M., JP25117530 and JP15H01337 to T.K.), Grants-in-Aid for Scientific Research (B) and (A) (MEXT KAKENHI, JP24390107, JP17H01544), JST CREST (JPMJCR19S5), Osaka City University (OCU) Strategic Research Grant 2018 for top priority researches to M.M. D.A. acknowledges a María de Maeztu Unit of Excellence grant MDM-2014-0435. Thanks are given to O. Conchillo and Prof. X. Daura for their enlightening suggestions and to Prof. P. Loewen for a careful reading of the manuscript. Many thanks are given to the XALOC beamline team at ALBA for their support during data collection and to the Crystallography Platform at the Barcelona Science Park (PCB), especially to Dr. Roman Bonet for his expert advice during SEC–MALS analysis. We also acknowledge to Dr. Sílvia Barceló from the Molecular Interactions Unit, IDIBELL and to Dr. Marta Taules from the Scientific and Technologic Department, UB for their technical support during SPR experiments. Finally, we are very grateful to Dr. Juan Delgado from the Immunology Department of the Hospital Universitari Parc Taulí for his helpful advices on performing ELISA analyses.

## Author contributions

Conceived and designed the experiments: D.V., I.F., T.K., K.N., M.M., and D.A. Performed the experiments: D.V., A.K., U.M., R.I., R.P.L., J.M., R.M., P.B., O.Q.P., M.P.S., T.K., T.A.K., and D.A. Analyzed the data: D.V., A.K., U.M., R.I., P.B., O.Q.P., M.E., I.S., J.E., M.F.H., M.P.S., J.P., A.S.F., M.L.S., S.M., K.S., T.K., T.A.K., I.F., and D.A. Contributed reagents/materials/analysis tools: A.S.F., M.E., K.N., M.M., and I.F. Wrote the paper: D.V., A.S.F., O.Q.P., M.L.S., T.K., M.M., I.F., and D.A.

## Competing interests

The authors declare no competing interests.
