## [Peer Review File · Nature Communications]

Reviewers' Comments:

Reviewer #1:

Remarks to the Author:

I have followed intently the requirement for P1 and P40/P90 in *M. pneumoniae* adherence and gliding motility since their original description 40+ years ago, and so for over 4 decades I have wondered about their structures, how they interact, and how they function in receptor binding. It is therefore very exciting to examine the findings of Vizarraga et al., which provide the first real such glimpse from a structural perspective. I cannot speak specifically to the details of the crystallography itself, but determining structures for P1 and P40/P90 will have an extremely high impact on the mycoplasma field and will be of interest for a wide range of other pathogens, both bacterial and viral, that bind to terminal sialic acid receptors. Moreover, the correlation with previous studies on epitope mapping and antigenic variation to position biologically important regions of P1 and P40/P90 on their 3-D structures adds great value to the study. But while this is beautiful and highly significant work, I do have several items of concern and question whether some conclusions reached are perhaps premature.

Major concerns:

1. Based on structures here, P40 is clearly a part of the P1 adhesion complex. And yet, as noted by the authors, there has been disagreement in the past whether this was indeed the case. P40 does not consistently associate with P1/P90, whether by chemical cross-linking or direct purification of adhesion complexes. Have the authors explored features from the structural analysis that could resolve this inconsistency? Perhaps this could be addressed in the manuscript.
2. The authors report dissociation constants in the micromolar range for P90 and 3-sialyllactose (3-SL) and 6-sialyllactose (6-SL) but do not address how these findings compare with published studies on the affinities / binding of these oligosaccharides by *M. pneumoniae*. Likewise, they fail to address why they observe similar dissociation constants for 3-SL and 6-SL, with the latter slightly lower, when others report that *M. pneumoniae* binds to 6-SL at a much lower affinity than to 3-SL (for example, Kasai et al., *J. Bacteriol.*, 2013; Williams et al., *Mol. Microbiol.* 2018).
3. The studies described in lines 318-360 appear to add very little value. It is already known that antibodies to certain parts of the C-terminus block adherence (e.g. Dallo et al., 1988). The more important question would seem to be why / how these antibodies are impacting the sialic acid-binding domain at the top of the P40/P90 "crown". Do these antibodies cause a conformational change in P1 that renders this domain no longer accessible? Why not start by making antibodies instead to the sialic acid-binding domain of P90 and see if they inhibit mycoplasma adherence?
4. The studies described in lines 362-389 likewise seem to add very little beyond the published work of Jacobs. In addition, it has long been recognized that P90 is an important immunogen in human infections (Leith et al., *J. Exp. Med.*, 1983).
5. Lines 397-399: "However, the present study demonstrates that the binding to sialic acid receptors relies on P40/P90." It was already known that P40/P90 are required for binding to sialic acid (Krause et al., 1982; Waldo, Jordan, and Krause, *J. Bacteriol.*, 2005; note that P40 and P90 were first discovered by 2-D page (Hansen et al., *Infect. Immun.* 1979) and designated C and B, respectively). Rather, the current study predicts that P40/P90 mediates sialic acid binding directly. While I do not disagree with that conclusion, it seems likely that this is not the final word on sialic acid binding by *M. pneumoniae*. Too many questions remain to conclude that the interactions seen with the purified protein directly reflect the sialic acid binding that occurs by intact cells. These include the unanswered question about how antibodies targeting P1, including its C-terminal region, inhibit adherence; the matter of affinity for 3-SL and 6-SL; the requirement for P30 in order for P1/P40/P90 to be functional in sialic acid binding; and the requirement that *M. pneumoniae* be metabolically active for cytoadherence to occur. In addition, do the high number of disordered domains in P1 limit the conclusions reached regarding what it cannot do in its purified state? Perhaps, for example, sialic acid binding is biphasic, with the domain on P90 mediating an initial interaction, with a subsequent conformational change in P1 (perhaps through its interaction with P30 or involving protein phosphorylation, for example), forming a second sialic acid binding site having a higher affinity for 3-SL than 6-SL and functioning in gliding motility.

Minor items:

1. L. 87: "M. pneumoniae mutants lacking this protein are non-infective." This statement is misleading. The study the authors cite examined 22 non-hemadsorbing mutants, 21 of which nevertheless still had P1. These other mutants lacked P40/P90, P30, or the HMW proteins, and their characterization led to the recognition that accessory proteins are required for P1 function.
2. Line 93: "dimer of...heterodimer" – this is confusing.
3. Line 100: reference 18 is the incorrect citation for this statement.
4. Lines 132-133: A statement regarding the significance of the R and RFree factors would be helpful, especially considering the breadth of the journal.
5. Line 180 and elsewhere: It would be helpful if the authors incorporated supplemental Fig. 1 into the main body of the paper (providing a schematic with amino acid numbering for P1 and P40/P90 and regions of interest referenced by their aa numbers in the text).
6. It would be helpful if the authors clarified what % of P40/P90 was crystalized (and perhaps indicated on the figure suggested in item #5).
7. Line 222: Did the authors look for potential Zn²⁺ binding sites?
8. Lines 235-238: The C-terminal domain of P40/P90 was predicted based on the corresponding region of P110, for which it has ~2/3 sequence identity. What degree / range of confidence is expected by this approach?

Duncan Krause

Reviewer #2:

Remarks to the Author:

The manuscript follows on from the previous work by Aparicio et al which examined the structure and assembly of the Nap adhesion complex from *Mycoplasma genitalium*. Here the authors take the investigation a step further, studying the related proteins P1 and P40/P90 from *Mycoplasma pneumoniae*. Given the sequence similarities between P1 and P40/P90 on the one hand, and P110 and P140 on the other, it is not too surprising that the folds and association of the heterodimers are very similar. This similarity extends to the sialic acid binding site on P40/P90, suggesting a functional similarity as well. The structural differences between P1 and P40/P90, as well as with their *M. genitalium* counterparts, are noted in some detail. Probably of more interest are the data on epitope mapping and sequence variability which form the latter part of the paper. The main conclusions seem to be that the C-terminal domain is more conserved and that antibodies against it block adhesion. Sera from infected individuals, selected for their reactivity against P1, also have IgG antibodies against P40/P90. Given previous publications, which the authors cite, these conclusions do not seem particularly novel. For example, the authors note the cross-reactivity between P1 and P140 (line 373). Taken together, these findings seem incremental on previous work.

Major comments

1. Omit maps and modelling of sialic acid containing ligands in the 3SL and 6SL complexes. The figures showing electron density for these complexes give few details (Fig 3C and Fig S8). What is the cutoff sigma value? Was only density around the ligand shown? At these low resolutions, modelling of the precise orientation of the ligands could be difficult.
2. Line 292 If there is no sialic acid binding site on P1, how is the cytoadherence mediated?
3. Line 302 'Immunodominant epitopes' is a rather imprecise term- presumably the authors mean B cell rather than T cell epitopes?
4. Line 305 What does 'quite buried' mean? it would help to clarify with accessible surface area calculations.
5. Line 316 and Fig 5 B/C The distinction between genetic and clinical variability is not very clear. Clinical isolates exhibit sequence variation, presumably- is that 'genetic' or 'clinical' or both?
6. Lines 347-350 These two sentences need re-writing: the point being made here is not clear.

7. Line 365 & 369 Seropositive patients were selected using the Liaison M. pneumonia IgG, IgM kit (DiaSorin). As I understand it, this detects IgG and IgM for reactivity against P1 antigen. It is not therefore surprising that most patients test positive for P1 reactivity (line 370) or, indeed, for P140, given their propensity to cause cross-reaction. What particular insight is there here?

Reviewer #3:

Remarks to the Author:

In their manuscript entitled: Immunodominant proteins P1 and P40/P90 from human pathogen *Mycoplasma pneumoniae*, the authors describe a crystal structure and cryo-EM structure of P1 and a crystal structure of P40/P90. In addition the authors solved crystal structures of P40/P90 in complex with two sialylated oligosaccharide fragments together with a few of immunological experiments.

The manuscript describes an impressive body of results from technically challenging experiments. The result section is well written and very clearly phrased. The discussion part appears quite speculative, though, and most of these speculations do not really enrich the manuscript. The novelty of many of the reported results is not entirely clear, either. In a previous publication, the authors already described the crystal structure of the P40/P90-homologous protein P110 from *Mycobacterium genitalium* in the presence and the absence of the identical sialylated oligosaccharides (Nat Commun. 2018 Oct 26;9(1):4471). P40/P90 and P110 share 44% sequence identity and the structures are highly similar. Also the sugar-binding mode is identical in both proteins.

In an accompanying manuscript that hasn't been published yet but that appears to have been already submitted, the authors describe the crystal structure of the P1-homologous protein P140 from *Mycobacterium genitalium* and of the P140/P110 complex as well as cryo-EM and -ET structures of the complex. Although technically not reporting the identical experiments, these three manuscripts together describe overlapping findings and hence share many conclusions, when taken into account the high sequence and structural homology between P1 and P140 as well as P110 and P40/P90.

Major points

The authors need to better describe how the phase problem has been overcome in the crystal structure determination of the P1 protein. The authors write that the structure was solved 'by averaging between crystals from both proteins despite neither molecular models nor experimental phases were available'. This is obviously not correct. Careful reading of the accompanying manuscripts suggests that initial electron density was obtained by cryo-EM. This density was then used for a molrep search and the resulting maps were then averaged across crystals and thereby the initial molrep phases were improved to the point where models could be built. If this is so, then this needs to be rephrased.

The authors performed size exclusion chromatography experiments and state that Fig. S6c shows that P1 and P40/P90 form heterodimers. This also holds true for P1 from *M. pneumoniae* and P110 from *M. genitalium*. Fig. S6 doesn't really show this since a single peak, as displayed in Fig. S6c and Fig. S6d, can also be observed when two proteins of similar size merely co-elute. The authors need to include/show chromatograms of the individual proteins together with chromatograms of the complexes.

The complexes of P40/P90N with S3L and S6L are of very limited resolution (3.1 Angs and 2.8 Angs resolution) and the experimental evidence that is provided to underline the correctness of the models is not entirely convincing. This is of course always a challenge at low resolution; therefore, considerable effort is required to corroborate and validate the proposed ligand interaction models.

Thus, the authors should display a simulated annealed omit map for this region for both complexes (and not only for 6SL, Fig. S8). The depiction of these maps should also include different orientations of the complexes. As it stands now, the final density is not very convincing for the refined complexes. Also, the interaction mode between ligand and protein is to some extent unexpected since intermolecular clashes appear to be present between ligand and protein in the 3.1 Angs structure at least. In a previous publication (Nat Commun. 2018 Oct 26;9(1):4471) the author described crystal structures of the homologous protein P110 from *M. genitalium* in complex with the same ligands S3L and S6L. These complexes were determined at 2.2 and 2.5 Angs resolution, respectively. However, also here, the electron densities for the ligands in the complexes, as retrieved from the PDB, are not really convincing.

My personal opinion is that it would be better at resolution as low as 3.1 and 2.8 Angs to not include/propose such detailed interaction models backed by such poor electron density. In my view, it would be perfectly fine if the authors would just state that they propose that sialylated oligosaccharides bind at this position based on unambiguous additional positive density that they observe when cocrystallizing P40/P90N with either S3L or S6L.

The authors should also consider generating single-site mutations followed by SPR measurements in order to validate the very detailed ligand-binding model that they propose.

Could the authors also comment on the reported SPR measurements? The authors used a biotinylated ligand for their SPR measurements. Are the crystal structures of the complexes in agreement with the accessibility of the ligand biotin tag required for the SPR measurements?

It would be good if the authors could show the SPR binding traces for the 3SL and 6SL ligands in Supplementary Fig. 9.

I have some difficulties with the immunological experiments/findings. First, the authors map existing binding epitopes onto the determined crystal structures. Secondly, the authors map genetic variability as discussed in the literature onto these structures. They then propose that the membrane-proximal and C-terminal domain of P1 'could be more effective in eliciting immunoprotection' than the N-terminal region. They then raised polyclonal antibodies against the C-terminal domain of P1 and investigated how these antibodies interfere with *M. pneumoniae* adhesion capabilities. In a last section, they investigate what kind of constructs from P1 and P40/P90 are recognized by sera from *M. pneumoniae* patients.

While these experiments are certainly interesting, in the context of the present manuscript they rather appear as add-ons. At the same time, they do not appear as being complete. Thus, instead of generating polyclonal antibodies only against the C-terminal domain, the authors should also investigate antibodies raised against the N-terminal region of P1 in order to more objectively and less circumstantially compare the immunoprotective potential of the C- versus the N-terminal domain. The investigations done with sera from patients clearly show that both the C- or the N-terminal domain are immunogenic and removing the N-terminal domain reduces recognition by about 74% with patient sera (if I read this correctly).

Minor points

On page 4, line 111, the authors write: ...we report the near atomic resolution structures... In contrast to atomic resolution, the term near atomic resolution doesn't have a fixed meaning. However, I am afraid, 1.9 Angs is really not near to atomic resolution, though.

The authors write on page 5 that they used Psipred for designing their construct. The authors might want to phrase/include two or three words about what the programme Psipred is actually predicting, namely secondary structure elements.

Page 7, line 201. How was the cleavage site determined. Why is this not included in the method section. The legend of Fig. S6 hints that Edman sequencing was used in case of some samples. Edman sequencing only provides N-terminal sequence. It doesn't actually show that the C-terminal residues are as stated by the authors. This should be better described.

Page 19, line 585. Please rephrase: 'with a pseudo-orthorhombic unit cell' to read 'pseudo-orthorhombic unit cell dimensions'.

The authors entitled Supplementary Fig. 2: 'Structural sequence alignments (with Esprit)'. As far as I understand, Esprit doesn't calculate any structural alignments and hence would not allow for deducing a structural sequence alignment. As far as I can see, Supplementary Fig. 2 shows the structurally annotated sequence of P1 to which the sequence of P140 from *M. genitalium* was subsequently aligned. If so, please correct this.

The comment from above also holds true for the title of Supplementary Fig. 5.

Supplementary Table I: Please add r.m.s. bond angle deviations and Ramachandran statistics in Table SI. Why are different geometry validation criteria listed in Table SI and Table SII? Please correct 'wavelength' in Table SI.

Reviewer #1:

I have followed intently the requirement for P1 and P40/P90 in *M. pneumoniae* adherence and gliding motility since their original description 40+ years ago, and so for over 4 decades I have wondered about their structures, how they interact, and how they function in receptor binding. It is therefore very exciting to examine the findings of Vizarraga et al., which provide the first real such glimpse from a structural perspective. I cannot speak specifically to the details of the crystallography itself, but determining structures for P1 and P40/P90 will have an extremely high impact on the mycoplasma field and will be of interest for a wide range of other pathogens, both bacterial and viral, that bind to terminal sialic acid receptors. Moreover, the correlation with previous studies on epitope mapping and antigenic variation to position biologically important regions of P1 and P40/P90 on their 3-D structures adds great value to the study.

We appreciate very much these words from the reviewer.

Major points

1. Based on structures here, P40 is clearly a part of the P1 adhesion complex. And yet, as noted by the authors, there has been disagreement in the past whether this was indeed the case. P40 does not consistently associate with P1/P90, whether by chemical cross-linking or direct purification of adhesion complexes. Have the authors explored features from the structural analysis that could resolve this inconsistency? Perhaps this could be addressed in the manuscript.

The reviewer is right; there was disagreement in the past whether P40 was part or not of the P1 adhesion complex. Although we cannot provide a definitive answer explaining the reasons for the disagreement, with the structural data now available, it is clear that P40 can be part of the Nap complex only if it remains bound to P90. Given the many interactions found between P40 and P90, which result in a highly stabilizing energy for the P40-P90 ensemble (according to the PISA analysis), it appears we should expect that P40 will remain bound to P90 in most circumstances (both *in vitro* and *in vivo*). However, as among the many interactions there are no covalent bonds between P40 and P90, it is always possible that under stringent or denaturing conditions, for example during certain purification or analysis protocols, P40 could become lost from the adhesion complex.

We have added two sentences in the Discussion section of the updated manuscript addressing this issue:

Line 406. “.....This is in agreement with the extensive interface between the P40 and P90 polypeptides indicating that they form a very stable ensemble (according to PISA³³) (Supplementary Figure 6). However, P40 is located at the outer surface part of the Nap and due to the absence of covalent bonds might become separated from the Nap complex under certain stringent or denaturing conditions. The loss of P40 only in certain conditions could explain the discrepancies in the past about whether or not P40 was part of the adhesion complex.....”

2. The authors report dissociation constants in the micromolar range for P90 and 3-sialyllactose (3-SL) and 6-sialyllactose (6-SL) but do not address how these findings compare with published studies on the affinities / binding of these oligosaccharides by *M. pneumoniae*. Likewise, they fail to address why they observe similar dissociation constants for 3-SL and 6-SL, with the latter slightly lower, when others report that *M. pneumoniae* binds to 6-SL at a much lower affinity than to 3-SL (for example, Kasai et al., *J. Bacteriol.*, 2013; Williams et al., *Mol. Microbiol.* 2018).

Affinities measured in our work correspond to *in vitro* analysis of the purified ectodomains from P1 and P40/P90. In turn, the published studies were mainly performed *in vivo* and consequently the whole Nap was involved in the binding process with other external factors such as interactions with P30 or the oligosaccharides surface density and distribution having an important role. We were doubtful about how far we could go with quantitative comparisons. In this sense, the discrepancies indicated by the reviewer, for the relative affinities of 3SL and 6SL, could be interpreted as due to the presence of P1 (in the Nap) and/or P30 modulating the binding of oligosaccharides to P40/P90, but other explanations might also be possible. In any case, we fully agree that it is important to have into account all the related information and the references indicated by the reviewer have been added in the updated manuscript as follow:

Line 393. “.....In addition to P1, other cytoadherence-accessory proteins had been reported as required for binding to sialic acids^{15,47}. The present study demonstrates that the binding *in vitro* to sialic acid receptors relies in fact on P40/P90 (**Figures 3c and Supplementary Figure 7**), however affinities and specificities could be tuned through other proteins or due to the density and distribution of the oligosaccharides in the surface^{38,39}.....”

Line 493. “.....Moreover, proteins others than the Nap components, including P30, P65, HMW1, HMW2 or PrpC/PrkC are known to be essential for binding and gliding⁵⁸⁻⁶¹. These proteins might support functioning and conformation of the Naps to play its roles..... ”

3. The studies described in lines 318-360 appear to add very little value. It is already known that antibodies to certain parts of the C-terminus block adherence (e.g. Dallo et al., 1988). The more important question would seem to be why / how these antibodies are impacting the sialic acid-binding domain at the top of the P40/P90 “crown”. Do these antibodies cause a conformational change in P1 that renders this domain no longer accessible? Why not start by making antibodies instead to the sialic acid-binding domain of P90 and see if they inhibit mycoplasma adherence?

It is true that previous work had already showed that antibodies raised against certain fragments of P1 can block adherence (as in Dallo et al., 1988) or detach gliding mycoplasma cells (as in Seto et al., 2005). We believe that those findings are complemented by our results as the structural information allows relating the level and properties of the immune response with very specific regions or domains of the protein. With the structural information it becomes evident that antibodies against just the C-domain of P1, away from the cell receptor binding site, can only indirectly interfere with adhesion and gliding. Our hypothesis, in line with the mechanism suggested by Seto *et al.*, contemplates the idea that these neutralizing antibodies against the C-terminal domain can hamper conformational changes required by the functioning of Naps, such as the ones reported recently in *M.genitalium* (Aparicio *et al* 2020. *Nat*

commun.). Structures of complexes with these antibodies might provide an answer about the blocking mechanism and we are evaluating the feasibility of this research. The structural information opened different venues for the work with antibodies. The sialic binding site in P40/P90 is surrounded by genetically and structurally variable regions and it appeared to us reasonable to prioritize the research with neutralizing antibodies towards the most constant parts of the adhesion complex.

4. The studies described in lines 362-389 likewise seem to add very little beyond the published work of Jacobs. In addition, it has long been recognized that P90 is an important immunogen in human infections (Leith et al., J. Exp. Med., 1983).

As the reviewer indicates, the work from Jacobs describes how IgGs and IgMs in sera of infected patients react against two octapeptides from the N-terminal domain of P1. Related with our reply of the previous point, we believe our results extend those findings by providing the structural framework with the crucial information about the accessibility of any peptide in the proteins. The structural data also opens the possibility of defining not just linear peptide epitopes but also consistent structural epitopes (such as the whole C-domain). The serology assays show now that the immune response, in terms of IgGs and IgMs, differs for different specific protein regions. We agree that full credit should be given to the work by Leith *et al.* determining (already in 1983!!) that P1 and P2 (P90) are potent immunogens of the human immune system when infected with *M.Pneumoniae*, and the text has been changed in the Discussion section trying to correct this.

Line 462. “.....P40/P90, often overlooked in immunological studies in spite of pioneering studies showing P90 was an important immunogen⁵¹, is recognized by 74% of the sera from infected patients.....”

No serologic data was reported for P40/P90 in that work or, to our knowledge, since then, and we believe a number of the results obtained now were quite unanticipated, such as the one indicating that insertion S2 (located within P40!) appears to contain strong immunogenic regions that might be acting as diverting decoys for the host immune system.

5. Lines 397-399: “However, the present study demonstrates that the binding to sialic acid receptors relies on P40/P90.” It was already known that P40/P90 are required for binding to sialic acid (Krause et al., 1982; Waldo, Jordan, and Krause, J. Bacteriol., 2005; note that P40 and P90 were first discovered by 2-D page (Hansen et al., Infect. Immun. 1979) and designated C and B, respectively). Rather, the current study predicts that P40/P90 mediates sialic acid binding directly. While I do not disagree with that conclusion, it seems likely that this is not the final word on sialic acid binding by *M. pneumoniae*. Too many questions remain to conclude that the interactions seen with the purified protein directly reflect the sialic acid binding that occurs by intact cells. These include the unanswered question about how antibodies targeting P1, including its C-terminal region, inhibit adherence; the matter of affinity for 3-SL and 6-SL; the requirement for P30 in order for P1/P40/P90 to be functional in sialic acid binding; and the requirement that *M. pneumoniae* be metabolically active for cytoadherence to occur. In addition, does the high number of disordered domains in P1 limit the conclusions reached regarding what it cannot do in its purified state? Perhaps, for example, sialic acid binding is biphasic, with

the domain on P90 mediating an initial interaction, with a subsequent conformational change in P1 (perhaps through its interaction with P30 or involving protein phosphorylation, for example), forming a second sialic acid binding site having a higher affinity for 3-SL than 6-SL and functioning in gliding motility.

We appreciate very much and mostly agree with the list presented by the reviewer with many of the important issues remaining about the functioning of Naps and about the binding to sialic acid by *M. pneumoniae*. The references indicated by the reviewer are now included in the manuscript in the sentence:

Line 393. “.....In addition to P1, other cytoadherence-accessory proteins had been reported to be required for binding to sialic acids ^{15,47}.....”

We also agree with the idea that sialic acid binding can be a “biphasic” process, initiated by P40/P90 and maybe later modulated by P1, with P30 and/or phosphorylation having also a significant role. For that reason we have included two sentences in the Discussion section pointing out that other proteins besides P40/P90 and P1 can also play a crucial role in gliding and adhesion:

Line 493 “.....Moreover, proteins other than the Nap components, including P30, P65, HMW1, HMW2 or PrpC/PrkC, are known to be essential for binding and gliding ⁵⁸⁻⁶¹. These proteins might support functioning and conformation of the Naps.....”

Minor items:

1. L. 87: “M. pneumoniae mutants lacking this protein are non-infective.” This statement is misleading. The study the authors cite examined 22 non-hemadsorbing mutants, 21 of which nevertheless still had P1. These other mutants lacked P40/P90, P30, or the HMW proteins, and their characterization led to the recognition that accessory proteins are required for P1 function.

The statement was misleading as the reviewer indicates and the sentence has been modified as follow:

Line 85. “.....Because all these relevant properties, P1 has been attracting attention since the late 1970s, although it was soon recognized that accessory proteins were also required for its functioning ¹⁵.....”

2. Line 93: “dimer of...heterodimer” – this is confusing.

The sentence has been modified as: “dimer of a P140-P110 complex”

3. Line 100: reference 18 is the incorrect citation for this statement.

The reviewer is right; the reference was not appropriated and has been replaced by:

Line 99. -Jacobs E, Pilatschek A, Gerstenecker B, Oberle K, Bredt W. “**Immunodominant epitopes of the adhesin of *Mycoplasma pneumoniae***”. J Clin Microbiol. 1990; **28**: 1194–1197.

-Atkinson, T.P., Balish, M.F. & Waites, K.B. **Epidemiology, clinical manifestations, pathogenesis and laboratory detection of *Mycoplasma pneumoniae* infections.** *FEMS Microbiol Rev* **32**, 956-73 (2008)

- Chourasia, B.K., Chaudhry, R. & Malhotra, P. **Delineation of immunodominant and cytadherence segment(s) of *Mycoplasma pneumoniae* P1 gene.** *BMC Microbiol* **14**, 108 (2014)

4. Lines 132-133: A statement regarding the significance of the R and Rfree factors would be helpful, especially considering the breadth of the journal.

We thank the reviewer for this suggestion that should facilitate reading. The R and Rfree factors are the most widely used criteria in X-ray crystallography to assess the agreement between a given structure and the corresponding diffraction data. Although these agreement factors present some conceptual and practical flaws, the long tradition in using them has overcome the limitations. For most well refined macromolecular structures the two R factors should have values of about between 17 and 25, with the Rfree being a few units higher than the R factor. The difference between the two values can be attributed to the bias introduced in the model during refinement and, in general, should be kept low. A statement (**Line 632**) has been added in the “Experimental Procedures” section explaining the significance of these factors.

5. Line 180 and elsewhere: It would be helpful if the authors incorporated supplemental Fig. 1 into the main body of the paper (providing a schematic with amino acid numbering for P1 and P40/P90 and regions of interest referenced by their aa numbers in the text).

As suggested by the reviewer we have incorporated the information from Supplemental Figure 1 in the main body of the paper as the new Figure 1c, adding also complementary information related with regions and residue numbers referenced in the text.

6. It would be helpful if the authors clarified what % of P40/P90 was crystallized (and perhaps indicated on the figure suggested in item #5).

88% is the percentage of the P40/P90 ectodomain that was crystallized. This information is now depicted in the new modified Figure 1c as suggested by the reviewer in the previous minor item #5.

7. Line 222: Did the authors look for potential Zn²⁺ binding sites?

We looked for potential Zn²⁺ binding sites in the structure of P40/P90 and some residues (mainly histidines) were considered as possible binding candidates. However, the amount of mutations required for a systematic analysis, together with the likelihood of misleading/unclear outcomes and the uncertainties about the candidates discouraged us from performing a mutational screening.

8. Lines 235-238: The C-terminal domain of P40/P90 was predicted based on the corresponding region of P110, for which it has ~2/3 sequence identity. What degree / range of confidence is expected by this approach?

According to a classical work (Chothia & Lesk 1986, “The relation between the divergence of sequence and structures in proteins” EMBO, 5: 823-6) the expected RMSD between two protein structures would be of 0.6-0.7 Å for a sequence identity of 60-70%. The work of Chothia & Lesk, now amply accepted, is included as a reference in a modified sentence in the updated manuscript:

Line 233. “.....The complete extracellular region of P40/P90 was generated by combining the structure of the N-terminal domain with the structure of the Pro1004-Pro1113 region of P40/P90 modeled according to the P110 C-terminal domain because the sequence identity is high (68%) and the expected RMSD low (~0.6 Å)³⁴”

Reviewer #2:

Major comments:

1. Omit maps and modelling of sialic acid containing ligands in the 3SL and 6SL complexes. The figures showing electron density for these complexes give few details (Fig 3C and Fig S8). What is the cutoff sigma value? Was only density around the ligand shown? At these low resolutions, modelling of the precise orientation of the ligands could be difficult.

The oligosaccharides omit maps, computed according to the “composite omit map” protocol in Phenix, are now depicted for both 3SL (in Figure 3c) and 6SL (in Supplementary Figure 7). The electron density around the oligosaccharides is represented at 0.9 sigma for both 3SL and 6SL. We agree with the reviewer that the quality of the maps (mainly due to the resolution available) does not allow defining accurately the binding interactions of the oligosaccharides with the protein and we have eliminated (as suggested) the panel showing possible hydrogen bonds and other interactions. We believe that, in spite of the non-optimal quality of the map, the presence of the oligosaccharides bound to P40/P90 is unquestionable. Having confirmed that the binding site in P40/P90 is fully consistent with the binding site in the orthologous protein P110 (Aparicio et al., 2018. Nat commun), we believe we can assume that the results obtained for mutants of residues from the binding site in P110 can also be applied to P40/P90.

2. Line 292 If there is no sialic acid binding site on P1, how is the cytoadherence mediated?

The binding site for the oligosaccharides cell receptors is located in P40/P90, which interacts with P1 in the Nap complex. Therefore, adherence is mainly due to P40/P90, although modulated by P1. This is a major result from our work, as for many years it was assumed things were different. In the manuscript we are trying to insist, both in the Results and in the Discussion sections, about the relevance and the novelty of this result.

3. Line 302 ‘Immunodominant epitopes’ is a rather imprecise term- presumably the authors mean B cell rather than T cell epitopes?

We were using “immunodominant epitopes” to indicate “dominant epitopes for antibody production” or maybe “immunodominant epitopes for B-cell immunity”. Trying to be clearer we have modified the text as:

Line 297. “Two other reported epitopes (Trp810-Tyr817 and Phe1124-Arg1131), recognized by sera from many of the *M. pneumoniae* infected patients ²¹, are located in the N-terminal domain (Figure 5a). The first of these exist inside the P1 molecule, while the second is exposed in the P1 surface, although its accessibility might also be limited in the Nap complex. These two epitopes might be immunodominant for antibody production; however antibodies against these epitopes may not have cytoadherence-inhibitory activities.”

4. Line 305 What does 'quite buried' mean? it would help to clarify with accessible surface area calculations.

As the reviewer indicates, 'buried' refers to a region in the structure with no surfaces accessible to solvent (in terms of computed accessible surface areas) and consequently a region that is somehow “buried” inside the structure. In this case, “quite buried” was trying to indicate (in a likely not very precise way) that the epitope was not exposed in the surface of the protein and consequently not accessible by antibodies, unless the protein is denatured or digested in small peptides. The text has been modified avoiding the expression “quite buried”

5. Line 316 and Fig 5 B/C The distinction between genetic and clinical variability is not very clear. Clinical isolates exhibit sequence variation, presumably- is that 'genetic' or 'clinical' or both?

The known clinical isolates exhibit variations that we refer to as “clinical variability”. On the other hand, “genetic variability” indicates regions that are potentially variable because they correspond to variable RepMP sequences, although this potential variability (which can be generated by intra-genome DNA recombination(s) among RepMPs) might not have (yet) been found in clinical isolates. This explanation is now included in the Figure 5 legend.

6. Lines 347-350 These two sentences need re-writing: the point being made here is not clear.

The sentences have been modified as:

Line 343. “As a final checking, the specificity of antibodies for the C-terminal constructs was also assessed. Serum from challenged rats presented no significant differences with their PPI when incubated with the corresponding constructs before the quantitative adhesion assay. “

7. Line 365 &369 Seropositive patients were selected using the Liaison M. pneumonia IgG, IgM kit (DiaSorin). As I understand it, this detects IgG and IgM for reactivity against P1 antigen. It is not therefore surprising that most patients test positive for P1 reactivity (line 370) or, indeed, for P140, given their propensity to cause cross-reaction. What particular insight is there here?

Reactivity with P1 was used mainly as a control. In our opinion, it was important to demonstrate that our recombinant proteins (P1 and others) can be detected by the sera using our immunoassay protocol, which is slightly different from that followed with the Liaison kit. Similarly, we agree that detection of P140 does not have novelty *per se*, but it puts our data in the context of previous immunological studies on *M. pneumoniae*.

Reviewer #3:

Major comments:

1. The authors need to better describe how the phase problem has been overcome in the crystal structure determination of the P1 protein. The authors write that the structure was solved 'by averaging between crystals from both proteins despite neither molecular models nor experimental phases were available'. This is obviously not correct. Careful reading of the accompanying manuscripts suggests that initial electron density was obtained by cryo-EM. This density was then used for a molrep search and the resulting maps were then averaged across crystals and thereby the initial molrep phases were improved to the point where models could be built. If this is so, then this needs to be rephrased.

We really thank the reviewer for raising this question after having also read the accompanying *M. genitalium* work (that the reviewer mentions). It is true that neither in the *M. genitalium* work nor in the *M. pneumoniae* manuscript as it was, the solution of the phase problem for P140 and P1 was described in detail. Now that the *M. genitalium* work can be referenced it is maybe possible/easier to offer a complete description of the procedure followed. Although, the approach to solve the phase problem for the P1 crystals is not conceptually new, we do not know of any other example where a protocol with similar steps has been successfully applied and consequently it might be methodologically interesting to report it. To this end we are changing the *M. pneumoniae* manuscript by adding a new section within "Experimental procedures" entitled:

Line 575. "Determination of the P1 crystal structure"

The crystals from P1 belong to space group C2 and could contain, by packing considerations, only one subunit in the asymmetric unit. We tried to solve these P1 crystals by heavy atoms and SeMet derivatives, but finally the resolution was achieved by density modification techniques applying solvent flattening and averaging with crystals from the *M. genitalium* orthologous protein P140, whose structure was also unknown at that time (Aparicio *et. al.*, 2020. Nat commun). With this procedure the structures of P1 and of P140 (with a sequence identity of 41% between their ectodomains) were solved simultaneously. The P140 containing crystals used corresponded to crystals from the complex of P140 with the N-domain of P110 (P140-P110N) and to crystals from P140 alone. Steps followed for the structure determination of P1 and P140 were the following:

i) A partial molecular replacement solution was obtained for the P140-P110N crystals using as searching model the N-domain from the P110 structure (Aparicio et al., 2018. Nat commun). This partial solution of the P140-P110N crystals contained four P110N subunits in the asymmetric unit. The four P110N subunits were organized as two pairs and within each pair the two subunits were related by accurate local (non-crystallographic) two-fold axis. In this partial solution no density was visible that could correspond to P140.

ii) From this partial solution of the P140-P110N crystals, a difference (Fo-Fc) map was generated at about 3 Å resolution. Positive densities in this difference map, expected to correspond mainly to P140, were tentatively reinforced by averaging (without applying any mask) using one of the local two-fold symmetries. This unmasked averaging also weakens density from other subunits in the crystal not related by the local two-fold symmetry. In this averaged density it was possible to differentiate continuous regions of strong and of weak density, but it was not yet possible to identify any consisting molecular feature from P140.

iii) The Cryo-electron Tomography (CET) map of a whole *M.genitalium* Nap (Aparicio et al., 2020. Nat commun) was then used to define an initial possible mask for P140. The CET map, at ~17 Å resolution, corresponded to a dimer of P140-P110 complexes, where the fitting of the P110 structures was unambiguous. The CET map was placed on the (Fo-Fc) averaged map by superposing the corresponding fitted P110 structures. Then, a putative “P140 initial map” was generated as the (Fo-Fc) averaged density inside the CET envelop corresponding to P140. After some polishing (avoiding overlapping with neighbor subunits or retaining regions with continuous densities) the assumed “P140 initial map” was used to perform a molecular replacement search (using program PHASER) obtaining a reasonable possible solution with four P110N structures and four “P140 initial maps”. Attempts to use the density from the CET maps directly for phasing, and not just as a mask, were unsuccessful in our hands.

IV) Iterative cycles of density averaging (and solvent flattening), alternating with manual readjustments of the P140 mask, were then performed with the P140-P110N crystals. The procedure converged giving clearer and more continuous density, although no secondary structures that could correspond to P140 were visible. Likely, the almost parallel orientation of the two non-crystallographic two-fold axes weakened the phasing power of this averaging. In spite of the limitations, the new density (within the updated mask) allowed obtaining a molecular replacement solution for the P1 crystals. Surprisingly, the new density did not provide a solution for the P140 alone crystals.

V) Density modification with averaging within the P140-P110N crystal and now also with the P1 crystals (using program DMMULTI. Cowtan et al; 1994) improved quickly the maps. The iterative cycles of DMMULTI were alternated with cycles of phase extension for the P1 crystals (using the *autobuild* protocol in Phenix at 1.94 Å resolution) and with manual readjustments of the averaging masks (that were updated about forty times). Model building

was carried forward for P1 and P140 in parallel. With about 50% of the structures traced, it was possible to obtain a molecular replacement solution for the P140 alone crystals that, with six subunits in the crystal asymmetric unit, facilitated the completion of the P1 and P140 structures.

P1 was then refined, giving agreement factors R and R_{free} of 18.7% and 22.9%, respectively (**Supplementary Table I**). The difference between these two factors, used traditionally to assess how well the available molecular model explains the experimental diffraction data. The difference between both factors is related with the bias introduced in the model during refinement. Despite the quality of the final map (**Figure 2b**) the P1 structure presented a significant amount of disordered residues located at the N end (missing residues 29-59) and in fourteen loops (102-105, 228-230, 259-268, 278-282, 298-300, 337-348, 831-847, 870-888, 923-928, 941-944, 1226-1232, 1308-1324, 1341-1349 and 1482-1495). The final refined structure has been deposited in the PDB with code 6RC9.

2. The authors performed size exclusion chromatography experiments and state that Fig. S6c shows that P1 and P40/P90 form heterodimers. This also holds true for P1 from *M. pneumonia* and P110 from *M. genitalium*. Fig. S6 doesn't really show this since a single peak, as displayed in Fig. S6c and Fig. S6d, can also be observed when two proteins of similar size merely co-elute. The authors need to include/show chromatograms of the individual proteins together with chromatograms of the complexes.

Although the estimated molecular weight of the single peak already suggested that it corresponded to the complex, we agree with the reviewer that these estimations might not be conclusive. The chromatogram corresponding to the P1-P40/P90 complex and the two proteins individually is attached in the new Supplementary figure S5 panel c, where a change in the elution volume can be observed in the Superdex 200 5/150 column. On the other hand for the chimeric P1-P110 the analysis shows only a minor change in the elution volume (likely related with the low stability of the complex). We feel it is better to remove the reference to the possibility of this chimeric complex in the updated text. If the reviewer is interested we will be pleased to provide the corresponding chromatogram of the P1-P110 complex.

3. The complexes of P40/P90N with S3L and S6L are of very limited resolution (3.1 Angs and 2.8 Angs resolution) and the experimental evidence that is provided to underline the correctness of the models is not entirely convincing. This is of course always a challenge at low resolution; therefore, considerable effort is required to corroborate and validate the proposed ligand interaction models. Thus, the authors should display a simulated annealed omit map for this region for both complexes (and not only for 6SL, Fig. S8). The depiction of these maps should also include different orientations of the complexes. As it stands now, the final density is not very convincing for the refined complexes. Also, the interaction mode between ligand and protein is to some extent unexpected since intermolecular clashes appear to be present between ligand and protein in the 3.1 Angs structure at least. In a previous publication (Nat Commun. 2018 Oct 26;9(1):4471) the author described crystal structures of the homologous protein P110 from *M. genitalium* in complex with the same ligands S3L and S6L. These complexes were determined at 2.2 and

2.5 Angs resolution, respectively. However, also here, the electron densities for the ligands in the complexes, as retrieved from the PDB, are not really convincing.

My personal opinion is that it would be better at resolution as low as 3.1 and 2.8 Angs to not include/propose such detailed interaction models backed by such poor electron density. In my view, it would be perfectly fine if the authors would just state that they propose that sialylated oligosaccharides bind at this position based on unambiguous additional positive density that they observe when cocrystallizing P40/P90N with either S3L or S6L.

The authors should also consider generating single-site mutations followed by SPR measurements in order to validate the very detailed ligand-binding model that they propose.

As pointed out by the reviewer, the structures of P40/P90 complexed with 3SL and 6SL have been solved at resolutions challenging an accurate definition of the interactions with the ligand. The partial occupancy of the oligosaccharides also contributes to weaken the density (that correlates with the high temperature factors observed for the oligosaccharides). We have eliminated (as suggested) the panel showing possible hydrogen bonds and other interactions. In the new figures, the oligosaccharides omit maps, computed according to the “composite omit map” protocol in Phenix, are now depicted for both 3SL (in Figure 3c) and 6SL (in Supplementary Figure 7). We believe that, in spite of the non-optimal quality of the map, the presence of the oligosaccharides bound to P40/P90 is unquestionable confirming that the binding site in P40/P90 is fully consistent with the binding site in the orthologous protein P110 from *M. genitalium* (Aparicio et al., 2018. Nat commun). Therefore, we believe we can assume that the results obtained for mutants of residues from the binding site in P110 can also be applied to P40/P90.

4. Could the authors also comment on the reported SPR measurements? The authors used a biotinylated ligand for their SPR measurements. Are the crystal structures of the complexes in agreement with the accessibility of the ligand biotin tag required for the SPR measurements?

SPR measurements were performed using the same ligands (3SL-PAA-biotin, and 6SL-PAA-biotin) than for *M.genitalium* (Aparicio et al., 2018. Nat commun). It is noteworthy to mention that the ligands used contains a long polyacrylamide chain connecting the biotinylated chemically linked tag with the oligosaccharides, which should avoid most steric clashes between biotin and the P1 and P40/P90 samples used. A sentence has been added in the experimental section trying to clarify this point (**Line 745**).

5. It would be good if the authors could show the SPR binding traces for the 3SL and 6SL ligands in Supplementary Fig. 9.

As suggested by the reviewer, the old Supplementary Figure 9 (renumbered as the new Supplementary Figure 8) includes now the sensorgrams of 3SL and 6SL binding.

6. I have some difficulties with the immunological experiments/findings. First, the authors map existing binding epitopes onto the determined crystal structures. Secondly, the authors map genetic variability as discussed in the literature onto these structures. They then propose that the membrane-proximal and C-terminal domain of P1 ‘could be more

effective in eliciting immunoprotection' then the N-terminal region. They then raised polyclonal antibodies against the C-terminal domain of P1 and investigated how these antibodies interfere with *M. pneumoniae* adhesion capabilities. In a last section, they investigate what kind of constructs from P1 and P40/P90 are recognized by sera from *M. pneumoniae* patients. While these experiments are certainly interesting, in the context of the present manuscript they rather appear as add-ons. At the same time, they do not appear as being complete. Thus, instead of generating polyclonal antibodies only against the C-terminal domain, the authors should also investigate antibodies raised against the N-terminal region of P1 in order to more objectively and less circumstantially compare the immunoprotective potential of the C- versus the N-terminal domain. The investigations done with sera from patients clearly show that both the C- or the N-terminal domain are immunogenic and removing the N-terminal domain reduces recognition by about 74% with patient sera (if I read this correctly).

In the framework of the structural information obtained and given the large amount of previous studies and efforts dedicated to understand the immunogenic properties of P1, we believe it was important to perform the experiments the reviewer refers to about the generation of polyclonal antibodies and about testing the patients response with specific constructs. However, as the reviewer also indicates, many other experiments with antibodies can be envisioned now that the structures are available. A large percentage of the adhesion complex surface corresponds to structurally or genetically N-terminal variable regions and we decided it might be interesting to prioritize research with neutralizing antibodies towards the most constant parts of this adhesion complex.

Minor points

1. On page 4, line 111, the authors write: ...we report the near atomic resolution structures... In contrast to atomic resolution, the term near atomic resolution doesn't have a fixed meaning. However, I am afraid, 1.9 Angs is really not near to atomic resolution, though.

The words "near atomic resolution" have been eliminated from this sentence.

2.

The authors write on page 5 that they used Psipred for designing their construct. The authors might want to phrase/include two or three words about what the program Psipred is actually predicting, namely secondary structure elements.

The text has been modified to include a brief explanation about Psipred:

Line 123. ".....spanning residue Met1 to Leu59 according to the structural properties prediction program Psipred²⁷....."

This software predicts, from the protein sequence, a diversity of molecular properties and structural features, such as disorder propensities and secondary structure elements or the location of transmembrane helices, signal peptides and the protein domains.

3. Page7, line 201. How was the cleavage site determined? Why is this not included in the

method section? The legend of Fig. S6 hints that Edman sequencing was used in case of some samples. Edman sequencing only provides N-terminal sequence. It doesn't actually show that the C-terminal residues are as stated by the authors. This should be better described.

As the reviewer indicates we have determined only the N-terminal sequences of the different bands. We believe that this information together with the approximate molecular weights of the corresponding bands provides a fairly complete description of the cleavage process taking place. The text, in the experimental procedures section, has been slightly modified describing briefly how the information was obtained.

4. Page 19, line 585. Please rephrase: 'with a pseudo-orthorhombic unit cell' to read 'pseudo-orthorhombic unit cell dimensions'.

Done

5. The authors entitled Supplementary Fig.2: 'Structural sequence alignments (with Esprit)'. As far as I understand, Esprit doesn't calculate any structural alignments and hence would not allow for deducing a structural sequence alignment. As far as I can see, Supplementary Fig.2 shows the structurally annotated sequence of P1 to which the sequence of P140 from *M. genitalium* was subsequently aligned. If so, please correct this.

We thank the reviewer for the correction. As the reviewer indicates, Esprit represents pre-aligned sequences that in our case were obtained by the structural superposition of P1 and P140 with the CCP4 software. The old Supplementary Figure 2 (renumbered as 1 in the new manuscript) has been modified accordingly.

6. The comment from above also holds true for the title of Supplementary Fig. 5.

The old Supplementary Figure 5 (renumbered as 4) has also been modified.

7. Supplementary Table I: Please add r.m.s. bond angle deviations and Ramachandran statistics in Table SI. Why are different geometry validation criteria listed in Table SI and Table SII? Please correct 'wavelength' in Table SI.

The information indicated by the reviewer has been added to Supplementary Table I and the geometry validation criteria unified between both Tables.

Reviewers' Comments:

Reviewer #3:

Remarks to the Author:

In the revised manuscript, the authors Vizarraga et al. addressed all the technical concerns that had been raised in the first reviewing round of their manuscript describing the structure of P1 and P40/P90 from *Mycoplasma pneumoniae*. In particular, the description of the structure determination process is now reported more clearly and also comprehensively. My major concern, however, remains the novelty of the reported findings.

These concerns appear even more valid now because a manuscript submitted in parallel by the same authors has by now been published in Nature Communications (Aparicio, D. et al., Nat Commun. 2020 Jun 8;11:2877. doi: 10.1038/s41467-020-16511-2.) This publication describes the structure determinations of highly homologous proteins, namely P110 and P140 from *Mycoplasma genitalium*. In addition, the description of the sialic acid-binding site in the homologous protein P110 from *M. genitalium* has been already published in an earlier manuscript (Aparicio, D. et al., Nat Commun. 2018 Oct 26;9(1):4471. doi: 10.1038/s41467-018-06963-y.) It is not clear to me how the findings of the present manuscript significantly extend beyond those now already published in two articles in Nature Communications.

This becomes even more clear if one considers that the significance of the body of work regarding immunogenicity and antibody epitope mapping that differs in these publications and that is presented as an add-on in the present manuscript has been questioned by several reviewers and including myself in the last reviewing round. Also, I have the impression that the criticisms raised regarding this body of work have only been addressed in parts in the rebuttal letter and the revision.

Minor points:

Line 215: replace Kcal by kcal

Reference 2 in the supplemental material section appears to be inadequate.

Subject: Manuscript final revision NCOMMS-20-16367A

Title: “Immunodominant proteins P1 and P40/P90 from human pathogen *Mycoplasma pneumoniae*” by Vizarraga *et al.*

Response to reviewers:

We would like to thank all reviewers for their positive comments on our manuscript.
Please see our response to reviewer #3

Reviewer #3

-Line 215: replace Kcal by kcal DONE!

-Reference 2 in the supplemental material section appears to be inadequate.

We thank the reviewer for this kind correction. The reference 2 in supplemental material is now properly assigned.